# Group testing via hypergraph factorization applied to COVID-19

David Hong [1], Rounak Dey [2], Xihong Lin [2,3,4 ✉], Brian Cleary [4 ✉] & Edgar Dobriban[1 ✉]

Large scale screening is a critical tool in the life sciences, but is often limited by reagents, samples, or cost. An important recent example is the challenge of achieving widespread COVID-19 testing in the face of substantial resource constraints. To tackle this challenge, screening methods must efficiently use testing resources. However, given the global nature of the pandemic, they must also be simple (to aid implementation) and flexible (to be tailored for each setting). Here we propose HYPER, a group testing method based on hypergraph factorization. We provide theoretical characterizations under a general statistical model, and carefully evaluate HYPER with alternatives proposed for COVID-19 under realistic simulations of epidemic spread and viral kinetics. We find that HYPER matches or outperforms the alternatives across a broad range of testing-constrained environments, while also being simpler and more flexible. We provide an online tool to aid lab implementation: http://hyper.covid19-analysis.org.

[1] Department of Statistics and Data Science, The Wharton School, University of Pennsylvania, Philadelphia, PA, USA. [2] Department of Biostatistics, Harvard T.H. Chan School of Public Health, Boston, MA, USA. [3] Department of Statistics, Harvard University, Cambridge, MA, USA. [4] Broad Institute of MIT and Harvard, Cambridge, MA 02142, USA. ✉email: xlin@hsph.harvard.edu; bcleary@broadinstitute.org; dobriban@wharton.upenn.edu

Biological screens that identify members of a large population with a disease have become invaluable tools for disease diagnosis and surveillance. When these screens are difficult to conduct or resources are limited, finding an efficient way to conduct the screen becomes critical. As such, widespread, scalable, and frequent testing is a defining challenge in combatting COVID-19 in the face of local, national, and global resource constraints. Pooled testing has recently arisen as a promising efficient scientific solution to the world-wide challenge of increasing COVID-19 testing capacity[1–17], encouraged in part by the finding that a single positive sample can be reliably detected by RT-qPCR in large pools[18].

The idea to test pools of samples dates back to the seminal work of Dorfman[19]. Dorfman testing is a two-stage approach where each individual is assigned to exactly one pool. In the first stage, pools are tested and each negative test result for a pool is applied to all its members. Only the remaining individuals (who are considered putative positives) are then individually tested in the second stage, which can greatly increase efficiency, depending on the pool size and prevalence of positive members of the population. A major strength of this approach is its simplicity (and thus robustness) in laboratory implementation; pools are easy to form and putative positives are simply the individuals in positive pools. Indeed, several early proposals[4–7] for COVID-19 pooled testing focus on Dorfman testing. However, it is well-known that Dorfman testing can have sub-optimal efficiency[9–12]; alternative designs use tests more efficiently and can thus screen more individuals, especially in the face of significant resource constraints.

There has been tremendous study and progress on pooled testing (also called group testing or specimen pooling) in general. Numerous works provide statistical[20–24], combinatorial[25–30], and information theoretic[31–44] perspectives, as well as software[45,46] to aid implementation, to name just a few. In addition, there has been a lot of work on analyzing and optimizing these methods for various constraints and evaluation criteria[47–57], often in the low prevalence regime. Broadly speaking, the approaches fall into three categories: (i) one-stage (or nonadaptive) approaches that identify positive individuals after only one round of pooled tests by using pools with carefully designed overlaps; (ii) two-stage[56–59] approaches (like Dorfman testing) that perform a first round of pooled tests to declare putative positives who are then individually tested in the second round; and (iii) multi-stage[59,60] (or adaptive/hierarchical) approaches that perform multiple rounds of pooled tests with pools chosen at each round based on the previous rounds.

Many recent works[8–10,12–16,61] focus on developing pooled testing methods for COVID-19. We will focus here on one-stage and two-stage approaches; multi-stage approaches can make robust lab implementation more difficult and can take longer to complete, which can make them less suitable for time-sensitive public health settings like COVID-19 testing. A leading one-stage method for COVID-19 is P-BEST[8], which splits each of 384 individuals into 48 partially overlapping pools and is designed for a prevalence around 1%. The pool assignments are based on a Reed-Solomon error-correcting code that enables identification from the single round of tests and provides robustness against, e.g., independent PCR failures. Positive individuals are identified by running a specialized decoding algorithm based on sparse regression. A leading two-stage method is plate-based array pooling[9], which arranges individuals into either an $8 \times 12$ or $16 \times 24$ grid (corresponding to plate sizes common in laboratory environments), then takes each column and each row to be a pool, resulting in 20 pools for 96 individuals or 40 pools for 384 individuals, respectively. Each individual is split into two pools and is a putative positive only if both the pools test positive. This approach retains some of the simplicity of Dorfman testing, while being potentially more efficient since individuals in only one positive pool do not need to be tested in stage 2. Overall, these pooling strategies can provide effective approaches for addressing the urgent, global need for efficient screening.

However, given the global nature of the pandemic, there are a wide variety of settings with differing needs and constraints, in which the proposed combinatorial designs may have limited utility[11]. P-BEST splits each sample into six pools, which can be time-consuming and error-prone to execute by hand, making it best-suited for well-resourced labs that have robotic-pipetting platforms. The specialized decoding algorithm used by P-BEST also adds complexity, making it more difficult to understand and implement without prior experience and expertise. Moreover, P-BEST and plate-based arrays (as well as many other proposed designs) are somewhat rigid and can be nontrivial to adapt to allow widely varying numbers of individuals screened per batch, available test kits, or prevalence of positive results. For COVID-19 screening in resource-limited settings, adapting to these various conditions is critical to achieving the greatest effectiveness[11]. Therefore, for this (and other) applications of pooled screening, there remains an outstanding need for a simple and flexible method that can be robustly implemented in diverse environments (without special equipment or expertise) and that can be easily tailored to optimize effectiveness (for diverse resource constraints).

We propose HYPER, a two-stage pooled testing method based on the combinatorics of hypergraph factorization. While the underlying mathematics is sophisticated, the resulting pools are simple to implement by hand (individuals are split at most three ways), and putative positives can easily be identified with only pencil and paper. We also provide an online tool (http://hyper.covid19-analysis.org) to facilitate implementation. The design accommodates any number of individuals while maintaining balance and efficiency. We characterize its behavior under a common statistical model and investigate its real-world COVID-19 performance through realistic simulations that model both viral kinetics and epidemic spread. HYPER outperforms both plate-based arrays and P-BEST in our experiments. These methods are particular instances of general array-based and code-based designs, so we also consider the broad classes of balanced arrays and Reed-Solomon Kautz-Singleton (RS-KS) code-based designs. In our experiments, HYPER also matches or outperforms these broad classes even in the scenarios where those classes excel in efficiency. For COVID-19 and beyond, HYPER represents a valuable addition to the growing toolbox for performing large-scale, pooled screens.

## Results

### The need for simple and flexible pooling designs that are balanced.
A pooling design is an assignment of each of $n$ individuals (or more generally, samples) to one or several of $m$ pools. We seek a simple and flexible pooling design that is balanced in the following natural ways (see the Supplementary Material for a formal definition):

(i) All individuals are assigned to the same number $q$ of pools; we focus on $q \le 3$ to aid lab implementation.

(ii) The $m$ pools are assigned as evenly as possible, i.e., the sizes of the pools are as close as possible to equal.

(iii) The $\binom{m}{q}$ possible pool combinations are assigned as evenly as possible.

Similar but nonidentical balance conditions have been widely studied in group testing (see the references above). The balance

conditions we consider here come from various naturally-arising real-life considerations that are especially relevant for COVID-19 testing. To begin, they all help make the pooling process more consistent, making robust implementation and quality control easier. For example, pool size determines how much volume to pipette from each individual in the pool. Balanced assignment of pools produces uniform pool sizes, making the volumes to pipette consistent across the design. Combined with a uniform assignment of all the individuals to $q$ pools, this also makes the volume needed from each individual consistent. Furthermore, a simple way to reduce pipetting steps in practice is to first pool individuals assigned to the same pool combination, then split this combined sample into the assigned pools. Balanced assignment of pool combinations makes the size of each of these combined samples consistent. Put together, these various forms of consistency simplify the pooling process and make it easier to add checks along the way, both of which help make it less error-prone.

Beyond making the pooling process more consistent, balance may also help make the performance of designs more consistent. For example, larger pools dilute positive samples more, which can increase the risk of false negatives. Balancing the assignment of pools can help make this reduction in sensitivity uniform across individuals. This is important in real-world testing; all individuals in the design should receive the same treatment. Similarly, balancing the assignment of pool combinations can help make efficiency more consistent by reducing the dependence of the stage 2 workload on which pool combinations test positive. This consistency in turn may help labs to plan so they can efficiently allocate tests. We study how balance impacts the consistency of both sensitivity and efficiency in more detail under the COVID-19 model below.

To summarize, real-world testing has a need for simple and flexible pooling designs that are also maximally balanced. Such designs aid robust lab implementation and encourage consistent performance across individuals. However, such designs turn out to be nontrivial to develop, and existing designs do not sufficiently address this aspect of real-world testing (see the Supplementary Material for more discussion). This paper fills the gap with HYPER, a simple and flexible method with maximally balanced pooling designs.

**HYPER pooling method.** We propose HYPER, a two-stage pooling strategy that uses maximally balanced pools. The first stage consists of pooled testing to identify putative positives, and the second stage consists of individually testing the putative positives. Individuals are assigned to pools in the first stage by cycling through a sequence of pool assignments obtained by solving a mathematical problem called hypergraph factorization. We explain the details here via a small example with $n = 12$ individuals each split into $q = 2$ out of $m = 6$ pools (Fig. 1).

The first step (Setup) is to obtain a sequence of pool assignments (AB, CD, etc.) via hypergraph factorization. As illustrated in Fig. 1, we think of the $m$ pools (A-F) as vertices (i.e., the black labeled points) and the $\binom{m}{q}$ possible pool assignments (AB, AC, etc.) as hyperedges (i.e., the blue lines connecting $q$ vertices each). A set of hyperedges (taken with the set of vertices) is called a hypergraph, and factorizing the hypergraph means partitioning the hyperedges (i.e., pool assignments) into subsets that each use all the vertices (i.e., pools) exactly once (see the Supplementary Material for more detail). In our example, this yields the five subsets shown as little hypergraphs within the circle of pool assignments: {AB, CD, EF}, {BC, DF, AE}, and so on. The setup step concludes by simply reading off the list to obtain

the sequence of pool assignments: AB, CD, EF, BC, and so on until AC.

The next step (Stage 1) is pooled testing to identify putative positives. First, place individuals in pools by cycling through the sequence of pool assignments, i.e., individual 1 is placed in pools A and B, individual 2 is placed in pools C and D, and so on. This yields a pooling design that assigns the $n = 12$ individuals to $q = 2$ of the $m = 6$ pools (which we denote as $H_{12,6,2}$). We do not use all $\binom{6}{2} = 15$ pool assignments in the sequence here since there are only 12 individuals. For more than 15 individuals, we would simply cycle through the sequence again until all the individuals were assigned. Next, we test the pools (pools B-D test positive in our example), and from these results, we identify putative positives in a process called decoding. For this step, HYPER uses conservative decoding, in which an individual is declared putative positive if it was in no negative pools. Namely, we eliminate all individuals in negative pools; the remaining individuals (who are in only positive pools) are the putative positives. Performing this elimination process in our example yields putative positive individuals 2, 4, and 7.

The final step (Stage 2) is individual testing of the putative positives. In our example, this means performing individual tests for individuals 2, 4, and 7. Individuals 4 and 7 test positive, so HYPER concludes by declaring them positive and declaring everyone else negative.

Note that the pooling design above is maximally balanced: each individual is in two pools, each pool contains four individuals, and each pool pair is assigned either once or never. Indeed, HYPER guarantees this balance in general by exploiting the properties of hypergraph factorization (see the Supplementary Material). Note that the HYPER pooling design is also flexible: it handles any number of individuals $n$ by simply cycling through the sequence obtained in the setup stage. Moreover, the hypergraph factorization needed to obtain that sequence can be efficiently constructed for $q = 1$ (for any $m$), for $q = 2$ (as long as $m$ is even), and for $q = 3$ (as long as $m$ is a multiple of six and $m - 1$ is a prime number); see the Supplementary Material for details. This covers a very wide range of useful design parameters $(n, m, q)$, and we provide an online tool (available at http://hyper.covid19-analysis.org) that generates pool assignments using these constructions, simplifying lab implementation for HYPER. The pools are also simple to implement in the lab and simple to decode, and the decoder can also be extended to correct for some false negatives (see the Supplementary Material). Table 1 summarizes these features of HYPER and compares with plate-based arrays[9] and P-BEST[8], as well as two existing random designs: random assignment[11] (i.e., assign each individual to $q$ pools independently and uniformly at random) and double-pooling[62] (i.e., partition the individuals into $m/2$ pools twice).

**Performance under a common statistical model.** We study the performance of HYPER under a common statistical model for group testing[19–24]. In this model, each individual (or in general contexts, each sample) is positive independently at random with probability $p$ (where $p$ is the disease prevalence) and the tests have independent errors. Namely, each test has a sensitivity of $\beta$ and a specificity of $1 - \alpha$, i.e., it returns positive with probability $\beta$ if it contains a positive individual and returns negative with probability $1 - \alpha$ if it contains no positive individuals. This model differs from the COVID-19 model we consider in the rest of the paper, but is also important to study since the potential applications of HYPER extend beyond COVID-19. Throughout this analysis, we further suppose that $n$ is a multiple of $m/q$ so that the pools are perfectly balanced with $k = nq/m$ individuals per pool. Note that this is not a significant restriction; for the

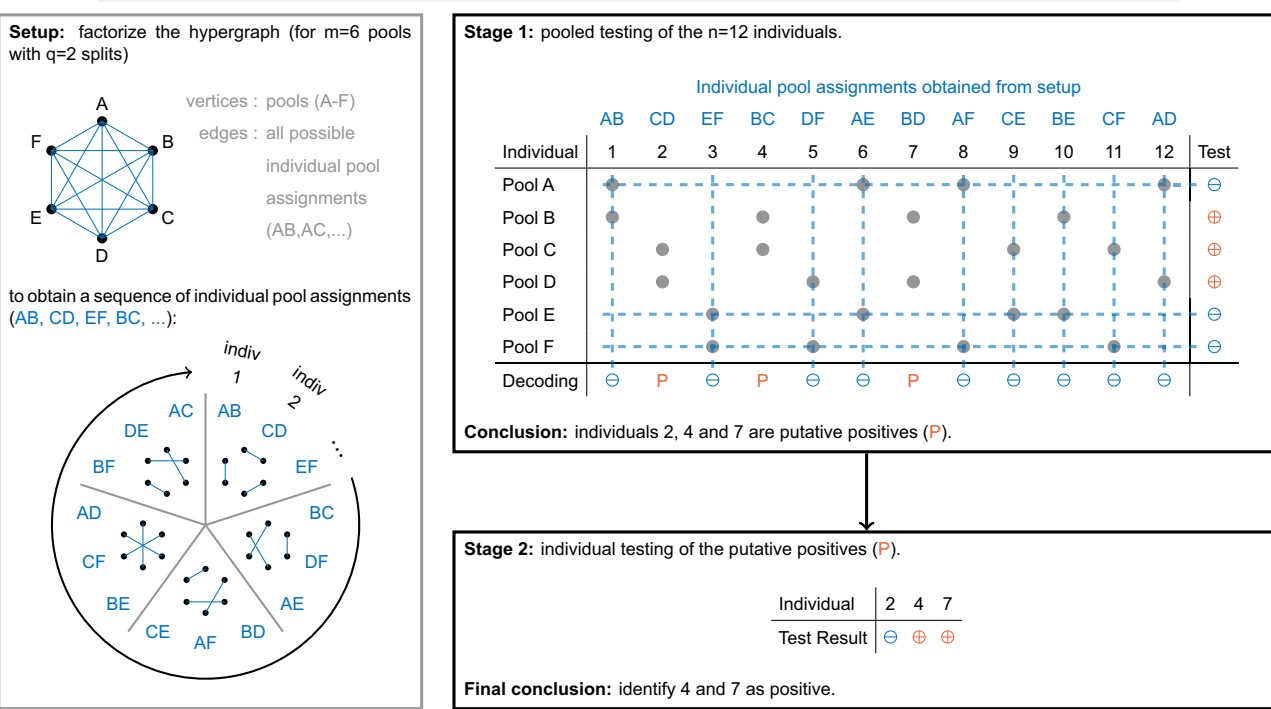

**Fig. 1 Illustration of HYPER.** Stage 1 tests pools that are formed by cycling through a sequence of pool assignments generated via hypergraph factorization. Putative positives are individuals that are not in any negative pools (decoding). Stage 2 tests the putative positives individually. In this example, $n = 12$ individuals (2 of whom are actual positives) are each split into $q = 2$ of $m = 6$ pools; three are decoded as putative positives and both positives are successfully identified in stage 2.

**Table 1 Comparison of various features of HYPER with existing methods.**

| | HYPER | Plate-based arrays[9] | | P-BEST[8] | Random assignment[11] | Double-pooling[63] |
| | | 8 × 12 | 16 × 24 | | | |
|---|---|---|---|---|---|---|
| # individuals per batch (n) | Any | 96 | 384 | 384 | Any | Multiple of m/2 |
| # pools (m) | Variable | 20 | 40 | 48 | Any | Even |
| # splits (q) | ≤ 3 | 2 | | 6 | Any | 2 |
| # stages | Two | Two | | One | Two | Two |
| Max. balanced pools | ✓ | × | | ✓ | × w.h.p.[a] | ✓ |
| Max. balanced combinations | ✓ | ✓ | | ✓ | × w.h.p.[a] | × w.h.p.[a] |
| Simple to implement by hand | ✓ | ✓ | | × | ✓ | ✓ |
| Flexible/easily adapted | ✓ | × | | × | ✓ | ✓ |
| Simple to decode by hand | ✓ | ✓ | | × | ✓ | ✓ |
| Corrects false positive | ✓ | ✓ | | ✓ | ✓ | ✓ |
| Corrects false negatives | Optional | × | | ✓ | Optional | × |

[a]With high probability, i.e., probability of failure ≫ 0.
In contrast to the existing methods, HYPER is simple to implement, flexible to adapt, and maximally balanced.

testing-constrained settings we are most interested in, the most effective designs often have large $n$ relative to $m/q$. Here, we present our key results (see the Supplementary Material for further details and derivations).

Our first result characterizes the expected number of tests $\mathbb{E}(T)$ used by HYPER (including the tests from stage 2). In particular, we have shown the following upper bound (which holds with equality when $n < \binom{m}{q}$ and $q \leq 2$):

$$\mathbb{E}(T) \leq m + n \cdot \left[ 1 - q p_1 + \binom{q}{2} p_2 \right], \tag{1}$$

where $p_1 = 1 - \beta + (\beta - \alpha) r^k$, $p_2 = p_1^2 + (\beta - \alpha)^2 r^{2k-u}(1 - r^u)$, $r = 1 - p$, and $u = \binom{m-2}{q-2} \cdot \lceil n / \binom{m}{q} \rceil$. In fact, we have derived a

sharper (but more complicated) version of this bound valid for all $q$ using the Dawson-Sankoff inequality[63,64] (see the Supplementary Material). We have also studied the overall accuracy of HYPER. In particular, for $q \leq 2$ and $n \leq \binom{m}{q}$, we have shown that the overall sensitivity and specificity are as follows:

$$\begin{aligned} \text{Sensitivity:} \quad & \Pr(\widehat{X}_i = 1 | X_i = 1) = \beta^{q+1}, \\ \text{Specificity:} \quad & \Pr(\widehat{X}_i = 0 | X_i = 0) = 1 - \alpha\gamma, \end{aligned} \tag{2}$$

where $X_i$ denotes the true status of individual $i$, $\widehat{X}_i$ is the status declared by HYPER, $\gamma = [\beta + (\alpha - \beta) \cdot r^{k-1}]^q$, and the probabilities are with respect to the random test errors and the random positivity of the other individuals. The corresponding false

negative and true positive probabilities (which can help guide how one interprets statuses declared by HYPER) are as follows:

$$\text{False negative probability:} \quad \Pr(X_i = 1 | \widehat{X}_i = 0) = \left(1 + o\,\frac{1-\alpha\gamma}{1-\beta^{d+1}}\right)^{-1},$$

$$\text{True positive probability:} \quad \Pr(X_i = 1 | \widehat{X}_i = 1) = \left(1 + o\,\frac{\alpha\gamma}{\beta^{d+1}}\right)^{-1},$$

$$(3)$$

where $o = (1 - p)/p$ is the odds ratio of prevalence. We illustrate some of these results with numerical simulations in Supplementary Fig. 1a to f. Notably, the sensitivity of HYPER under this model is independent of the pool sizes (in contrast to the COVID-19 model below). While it is beyond our present scope to do a thorough comparison, we note that corresponding calculations of sensitivity and specificity under this model have also been reported for array methods[21]. Note also that the sensitivity of HYPER under this model may be improved via error-correction of false negatives in stage 1 (see the Supplementary Material). However, doing so can come at the cost of efficiency, and analyzing this tradeoff is also beyond our present scope.

Focusing on the noiseless setting, i.e., test sensitivity and specificity close to 1 ($\alpha = 0$, $\beta = 1$), enables some further investigation. We first consider choosing optimal HYPER design parameters for large batches ($n \to \infty$) with diminishing prevalence ($p \to 0$). In this regime, we show that the optimal number of pools per individual $m/n$ and the corresponding expected tests per individual $\mathbb{E}(T)/n$ for HYPER with $q = 2$ are approximately (Supplementary Fig. 1g, h)

$$m/n \approx 2p^{2/3} - p, \quad \mathbb{E}(T)/n \approx 3p^{2/3}.$$

This improves upon Dorfman testing, for which the optimal expected tests per individual is approximately $\mathbb{E}(T)/n \approx 2p^{1/2}$ in this regime[20], and it matches that of three-stage Dorfman testing[20,62]. Designs with better efficiency in this regime are available in the literature, but they typically rely on using multiple stages[10,20] or taking $q$ much larger[47–54], each of which is outside our constraints (see the Supplementary Material for more discussion).

In the context of widely-spread infectious disease, it is also important to consider fixed (non-diminishing) prevalence $p > 0$, i.e., the linear regime[34–36]. Recent works in this regime have considered two problem formulations where group testing increases efficiency. The first problem is to maximize efficiency using any group testing method (i.e., any number of stages, any decoder, etc.). The second problem adds some real-life constraints by instead allowing only two-stage group testing methods with conservative decoding[36], such as array methods and HYPER. We compared known lower bounds for these two problems with the upper bound for HYPER given above, with $m$ and $q$ numerically optimized for $n = 6144$ (Supplementary Fig. 2). As one would naturally expect, better efficiency is achievable for problem 1, e.g., by using fully-adaptive methods[35], since it is much less constrained than problem 2. For problem 2, which incorporates real-life constraints, HYPER appears to be somewhat close to optimal.

To summarize, these results characterize the expected efficiency and accuracy of HYPER under a standard statistical model. The model captures important features of many applications beyond COVID-19 testing and provides a useful setting to evaluate HYPER. We found that for noiseless tests, the efficiency of HYPER for diminishing prevalence is competitive with other existing methods of comparable simplicity. For two-stage conservative testing with non-diminishing prevalence, HYPER appears to be somewhat close to optimal.

**Performance under a COVID-19 model**. We study the performance of HYPER under the viral load based COVID-19 model of Cleary and Hay et al.[11] It simulates: (a) SARS-CoV-2 viral load kinetics in infected individuals; (b) the dilution of viral loads during pooling that may lead to false negatives; and (c) the evolution of infection prevalence in a large population over time during epidemic growth and decline. We focus here on a window during which the infection prevalence (i.e., the percentage of individuals with nonzero viral load) increases exponentially from 0.03% to 2.46% (days 40–90 in our simulation) and individual testing has a sensitivity of roughly 85% (Fig. 2).

We compare $q = 2$ HYPER designs with Dorfman pooling (i.e., $q = 1$ HYPER designs) and two leading proposals: plate-based arrays[9] and P-BEST[8]. These methods use batches of $n = 96$ individuals ($8 \times 12$ array; Fig. 2a) or $n = 384$ individuals ($16 \times 24$ array and P-BEST; Fig. 2b). For each method, we consider the efficiency relative to individual testing (i.e., the number of individuals screened divided by the average number of tests used, including any stage 2 tests) and the average sensitivity (i.e., the percentage of positive individuals correctly identified) for each day in the simulation.

For $n = 96$ (Fig. 2a), we compare the $8 \times 12$ array with a $q = 2$ HYPER design ($H_{96,16,2}$) and a Dorfman design ($H_{96,8,1}$), both chosen to dilute samples a similar amount as the array and thus have potentially similar sensitivity. Our simulation shows that all three methods indeed have similar sensitivity, all roughly 10 percentage points lower than that of individual testing (Fig. 2a, bottom panel) due to the dilution of viral loads below the limit of detection in pooled testing. For much of the 50-day window, the array is roughly 4.8 times more efficient than individual testing, while $H_{96,16,2}$ is roughly 6 times more efficient than individual testing (Fig. 2a, top panel). In other words, $H_{96,16,2}$ is 25% more efficient than the array with essentially the same sensitivity. The Dorfman design is initially even more efficient but its efficiency significantly degrades with increasing prevalence; the $q = 2$ HYPER design is more efficient once the prevalence exceeds roughly 1%.

For $n = 384$ (Fig. 2b), we compare the $16 \times 24$ array and P-BEST with a $q = 2$ HYPER design ($H_{384,32,2}$) and a Dorfman design ($H_{384,16,1}$), both again chosen to dilute samples a similar amount as the array. The $q = 2$ HYPER design is again roughly 25% more efficient than the array design (Fig. 2b, top panel) while having essentially equal sensitivity (Fig. 2b, bottom panel). Likewise, it is again more efficient than the Dorfman design once the prevalence exceeds roughly 0.2%. In contrast to these two-stage methods, the one-stage approach of P-BEST has a constant efficiency of 8 times individual testing (Fig. 2b, top panel), but it significantly loses sensitivity around day 80 as prevalence grows (Fig. 2b, bottom panel). This is because the design and decoding algorithm are optimized for a prevalence around 1% and performance degrades beyond this operating point. For the two-stage methods, sensitivity instead increases. Notably, error-correcting does not appear to effectively handle the false negatives that arise here due to diluted viral loads falling below the limit of detection. P-BEST is generally the least sensitive among the pooling strategies.

Since the flexibility of HYPER allows for many designs, we next compared different HYPER designs and their various tradeoffs. Specifically, we considered various choices for the number of pools (Fig. 2c, $m = 32, 16, 12$) and the number of splits (Fig. 2d, $q = 1, 2, 3$). Similar to earlier studies of random assignment designs[11], the HYPER designs with a smaller number of pools $m$ are generally more efficient (especially when the prevalence is small) but slightly less sensitive. Likewise, designs with a larger number of splits $q$ are more robust to increasing prevalence (they do not lose as much efficiency) but they also tend to be less

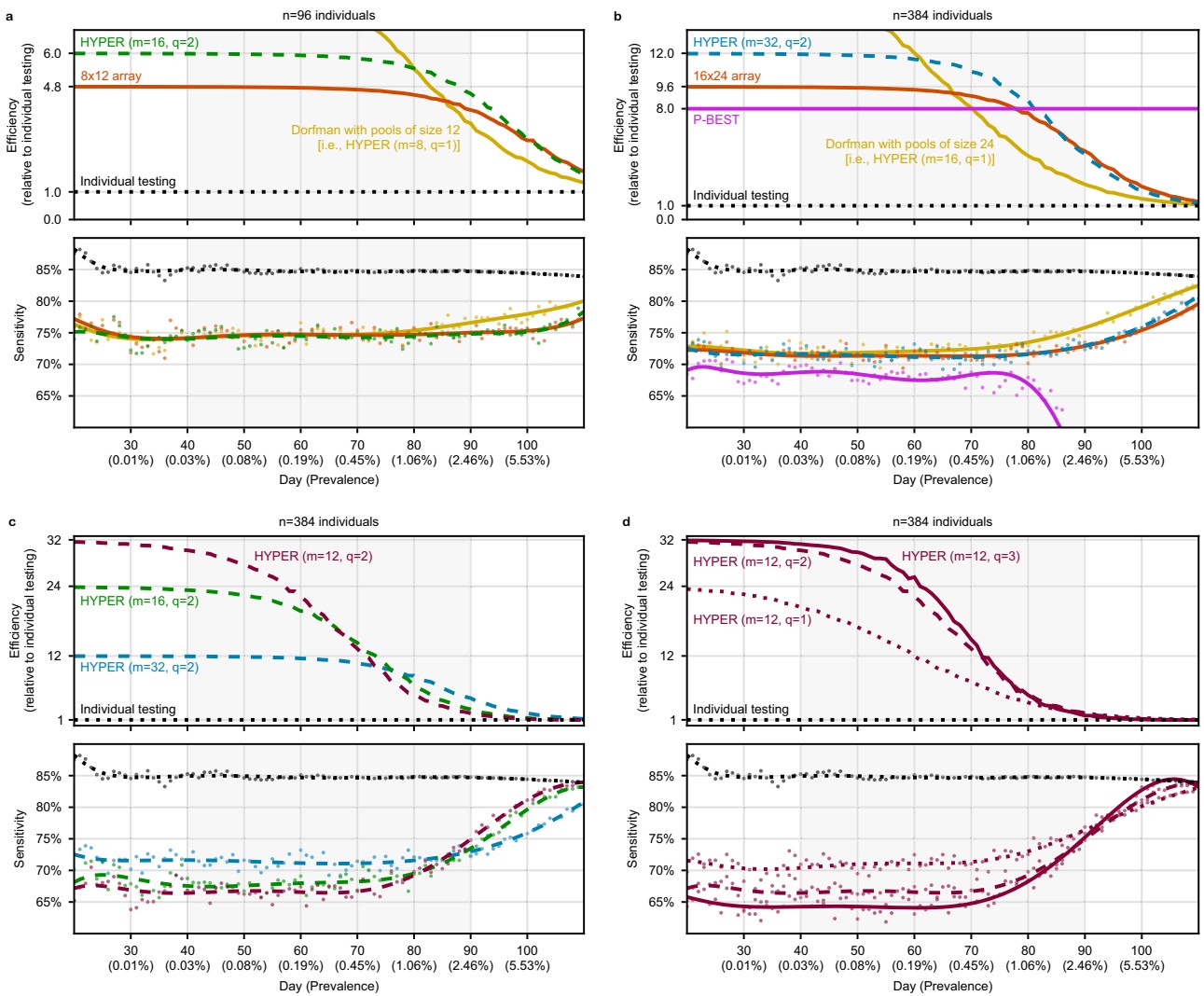

**Fig. 2 Efficiency and sensitivity of pooled testing during a simulated epidemic.** Average values of efficiency (relative to individual testing) and sensitivity of a variety of pooling designs are shown for each day, with results averaged across 200,000 random trials. For sensitivity, raw averages are shown as dots with degree-8 polynomial fits overlaid as curves; the curves for efficiency depict raw averages. During the days 40–90 (highlighted), the prevalence grows exponentially from 0.03% to 2.46%. **a, b** Comparison of HYPER with alternative methods that use $n = 96$ individuals per batch (**a**) or $n = 384$ individuals per batch (**b**). HYPER designs with $q = 2$ splits were chosen to have the same maximum pool sizes ($nq/m = 12$ for $H_{96,16,2}$; $nq/m = 24$ for $H_{384,32,2}$) as the array designs. Dorfman designs (i.e., HYPER designs with $q = 1$) with matching pool sizes are also included. Sensitivity (bottom panels) depends heavily on pool sizes, due to dilution of viral loads. **c, d** HYPER evaluated with varying numbers of pools (**c**, $m = 32, 16, 12$) and numbers of splits (**d**, $q = 1, 2, 3$). The designs are affected by the increasing prevalence over time to varying degrees. As prevalence increases, efficiency decreases (as more stage 2 tests become necessary), while sensitivity increases (as larger viral loads begin to rescue small viral loads that would have been missed). More efficient designs tend to be less sensitive, creating a trade-off.

sensitive. Overall, more efficient designs tend to be less sensitive, creating a trade-off that depends significantly on prevalence.

We also expanded the above comparisons in a few ways. First, we considered HYPER designs that match the number of pools in the plate-based arrays (Supplementary Fig. 3); these designs had essentially the same efficiency as their plate-based array counterparts but slightly higher sensitivity. Similarly, we considered HYPER designs that match the number of pools and the pool sizes of P-BEST (Supplementary Fig. 4). Matching the number of pools yielded similar efficiency to P-BEST at low prevalence but better sensitivity, while matching the pool sizes yielded similar sensitivity at low prevalence but better efficiency. We also compared HYPER with additional methods, beginning with balanced variants of the plate-based arrays (square arrays with holes), random assignment[11], and double-pooling[62] (Supplementary Fig. 5). The balanced arrays have similar efficiency but

slightly higher sensitivity than their plate-based array counterparts, so HYPER is again roughly 25% more efficient but is now slightly less sensitive. Random assignment and double-pooling have similar average performance to their corresponding HYPER designs. However, as discussed below, their performance can be more inconsistent. Since plate-based arrays and P-BEST are particular instances of general array-based and code-based designs, we also considered more general balanced arrays (that place multiple individuals in some array cells) and Reed-Solomon Kautz-Singleton (RS-KS) code-based designs (Supplementary Fig. 6). Compared to both these methods, HYPER had similar or better performance, with greater improvements in efficiency for more aggressive designs that use fewer pools and yielded greater efficiency at low prevalence.

Finally, we investigated how the balance of HYPER designs impacts the consistency of performance across individuals. In

particular, we studied how the sensitivity and efficiency achieved for a single positive individual varies as a function of where that individual is placed in the design (Supplementary Fig. 7). We compared HYPER with random assignment, double-pooling, consecutive pooling, lexicographic pooling, balanced arrays, and RS-KS codes. Overall, the designs with balanced pools (double-pooling, consecutive pooling, balanced arrays, RS-KS codes, and HYPER) had uniform sensitivity. Designs with imbalanced pools use varying volumes from each individual and dilute their viral loads to varying degrees, which can result in uneven sensitivity. Similarly, the designs with balanced pool combinations (lexicographic pooling and HYPER) had uniform efficiency. Consecutive pooling had uniform but lower efficiency; it uses only a subset of the possible pool combinations but does so in a balanced way. HYPER, which has perfectly balanced pools and pool combinations, had uniform sensitivity and efficiency. Moreover, its median efficiency (5.68 individuals/test) and median sensitivity (74.4%) were generally among the best.

**Choosing a pooling method given resource constraints.** In practice, decision makers must often choose a pooling method given limited resources for daily testing and sample collection. One approach is to maximize the number of individuals screened per day, i.e., the number of individuals $n$ per batch times the number of batches $b$ that can be run per day. However, while this metric accounts for the impact of resource constraints, it does not represent the actual number of infected individuals that the population screen can identify. A very efficient method could screen numerous individuals but still miss all the infected ones if it is not also sensitive.

Thus, we instead consider maximizing the number of individuals screened times the average sensitivity, i.e., an effective number of individuals screened per day, which we call the "effective screening capacity". Specifically, we study the problem of maximizing the effective screening capacity across days 40–90 of the above COVID-19 simulation (Fig. 2), given a range of resource constraints (limited amount of sample collection and testing). A nice property of the effective screening capacity is that scaling it by prevalence measures the average number of infected individuals per day that are identified by the screen. This makes it especially meaningful to maximize in public health contexts, where the goal may be to find and isolate as many infected individuals as possible. Here, we compare HYPER (optimized over a sweep of design parameters; Table 2) with individual testing, plate-based arrays[9] (optimized across the two configurations), and P-BEST[8]. We will first consider a few specific scenarios to understand the tradeoffs with each method, then consider a larger grid to get a picture of the overall trends.

We first consider a testing-scarce setting (Fig. 3a) with an average testing budget of 12 tests per day that is far outstripped by an average sample collection budget of 3072 samples per day. In this case, individual testing only screens 12 individuals and achieves an effective screening capacity of 10.2 individuals per day (the average sensitivity of individual testing is 84.8% since some positive individuals have viral load below the limit of detection). In contrast, the best HYPER design here ($H_{192,6,2}$) achieves an effective screening capacity of 122.2 individuals per day, roughly 12 times that of individual testing. It does so by pooling $n = 192$ individuals per batch into $q = 2$ of $m = 6$ pools with an average of $b \approx 0.9$ batches run per day (recall that some of the testing budget is used by stage 2 tests). Both plate-based arrays and P-BEST use more than 12 tests in a single run so do not satisfy the testing constraints here.

As the testing budget grows to 24 then 48 tests (Fig. 3b, c) with the sample budget unchanged, larger effective screening capacities

become possible by using larger designs, including the $8 \times 12$ array followed by the $16 \times 24$ array and P-BEST. HYPER adapts to these settings as well, and the larger designs here are accompanied with a larger number of splits $q$. HYPER remains the most effective overall, achieving effective screening capacities ~15 times that of individual testing and ~3 times those of the plate-based arrays and P-BEST.

When the testing budget grows to 768 tests per day (Fig. 3d), i.e., one-fourth of the sample collection budget, the pooled testing methods remain more effective than individual testing, but now by less than 4 times. In this increasingly testing-rich regime, P-BEST and the plate-based arrays are sample-constrained and under-utilize testing resources. P-BEST uses only $mb = 384$ of the 768 available tests, since all 3072 available samples are tested after $b = 8$ batches of $n = 384$ samples. The same is true for the $16 \times 24$ array design, although additional tests are used in stage 2. The most effective HYPER design $H_{6,1,1}$ corresponds to simple Dorfman testing, uses roughly 508 tests in the first stage, and achieves an effective screening capacity of 2375.1 individuals per day.

Finally, we consider two settings well-suited for plate-based arrays and P-BEST: 96 samples with 24 tests (Fig. 3e) for which the $8 \times 12$ array is well-suited (recall that some of the testing budget is used by stage 2 tests), and 384 samples with 48 tests (Fig. 3f) for which the $16 \times 24$ array and P-BEST are well-suited. Namely, these are settings where the sampling and testing budgets are close to the number of individuals and tests used by these designs. This can help them maximally utilize both the testing and sample collection budgets, i.e., neither resource is under-utilized. The plate-based arrays and P-BEST performed similarly to HYPER in these favorable cases, but notably HYPER remained slightly more effective: effective screening capacities of 74.1 vs. 71.1 for the first scenario and 265.5 vs. 262.2 for the second scenario.

Expanding this analysis to a grid of sampling and testing budgets gives a broad view of overall trends. We consider a sweep with each resource budget ranging from 12 to 6144 (Fig. 3g). Note first that for any given sample collection budget, the effective screening capacity grows as the testing budget scales up until it matches or outpaces sample collection. Individually testing all samples collected is most effective from that point on. Likewise, for any given testing budget, the effective screening capacity rises as the sample collection budget grows, eventually reaching an upper limit at which point testing becomes the limiting factor. Overall, pooled testing increases this upper limit, enabling an effective screening capacity far beyond the actual number of available tests. For example, for a testing budget of 96 tests per day, pooled testing achieves an effective screening capacity of up to 1500.9 individuals per day, which is over 18 times the effective screening capacity of 81.4 individuals per day achievable by individual testing.

Across the testing-constrained regime, i.e., where the testing budget is less than the sample collection budget, HYPER outperforms both plate-based arrays and P-BEST (Fig. 3 and Supplementary Fig. 10). Notably, Dorfman testing (i.e., HYPER with $q = 1$) is most effective when the testing budget is within a quarter of the sample budget. However, as the sample budget begins to further outstrip testing, combinatorial designs that involve more individuals $n$ per batch and that use more splits $q$ become most effective, consistent with earlier studies of analogous random designs[11].

The above results consider the effective screening capacity of each method across a 50-day window of epidemic spread. We next investigated the effective screening capacity on individual days, each corresponding to a different fixed prevalence. At low to moderate prevalence of 0.1% (Supplementary Fig. 11), 1.06%

**Table 2 List of HYPER designs $H_{n,m,q}$ considered.**

**q = 1**

| n | m | | n | m | |
|---|---|---|---|---|---|
| 4 | 1 | | 384 | 1 | |
| 8 | 1 | | 384 | 2 | |
| 12 | 6 | * | 384 | 3 | |
| 20 | 10 | | 384 | 4 | |
| 40 | 10 | | 384 | 6 | * |
| 40 | 20 | | 384 | 8 | |
| 44 | 11 | | 384 | 12 | |
| 44 | 22 | | 384 | 16 | |
| 48 | 12 | | 384 | 24 | |
| 48 | 16 | | 384 | 32 | |
| 48 | 24 | | 384 | 48 | |
| 64 | 8 | | 384 | 64 | |
| 64 | 16 | | 384 | 96 | |
| 64 | 32 | | 384 | 128 | |
| 80 | 20 | | 384 | 192 | |
| 96 | 2 | | 400 | 2 | |
| 96 | 3 | | 400 | 10 | |
| 96 | 4 | | 400 | 20 | |
| 96 | 6 | | 400 | 40 | |
| 96 | 8 | | 400 | 50 | |
| 96 | 12 | | 400 | 80 | |
| 96 | 16 | | 400 | 100 | |
| 96 | 24 | | 400 | 200 | |
| 96 | 32 | | | | |
| 96 | 40 | | | | |
| 100 | 20 | | | | |
| 192 | 12 | | | | |
| 192 | 16 | | | | |
| 192 | 32 | | | | |
| 192 | 48 | | | | |
| 192 | 64 | | | | |
| 192 | 96 | | | | |
| 288 | 12 | | | | |
| 288 | 18 | | | | |
| 288 | 36 | | | | |
| 288 | 48 | | | | |
| 288 | 96 | | | | |
| 288 | 144 | | | | |
| 360 | 40 | | | | |

**q = 2**

| n | m | | n | m | |
|---|---|---|---|---|---|
| 12 | 6 | | 384 | 42 | |
| 12 | 12 | | 384 | 44 | |
| 20 | 12 | | 384 | 46 | |
| 20 | 20 | | 384 | 48 | * |
| 24 | 6 | | 384 | 64 | * |
| 24 | 12 | | 384 | 96 | |
| 40 | 12 | | 384 | 192 | |
| 40 | 20 | | 384 | 288 | |
| 40 | 40 | | 400 | 12 | |
| 44 | 12 | | 400 | 20 | |
| 44 | 20 | | 400 | 40 | |
| 44 | 40 | | 400 | 44 | |
| 44 | 44 | | 400 | 48 | |
| 48 | 6 | | 400 | 64 | * |
| 48 | 12 | | 400 | 96 | |
| 48 | 20 | | 400 | 192 | |
| 48 | 24 | | 400 | 288 | |
| 48 | 40 | | 400 | 384 | |
| 48 | 44 | | 768 | 12 | |
| 48 | 48 | | 768 | 24 | |
| 64 | 12 | | 768 | 48 | |
| 64 | 20 | | 768 | 96 | |
| 64 | 40 | | 1536 | 24 | |
| 64 | 44 | | 1536 | 48 | |
| 64 | 48 | | 1536 | 96 | |
| 64 | 64 | * | 3072 | 48 | |
| 96 | 4 | * | 3072 | 96 | |
| 96 | 6 | | 6144 | 96 | |
| 96 | 8 | * | | | |
| 96 | 10 | | | | |
| 96 | 12 | | | | |
| 96 | 14 | * | | | |
| 96 | 16 | | | | |
| 96 | 18 | | | | |
| 96 | 20 | * | | | |
| 96 | 22 | | | | |
| 96 | 24 | | | | |
| 96 | 26 | * | | | |
| 96 | 28 | | | | |
| 96 | 30 | | | | |
| 96 | 32 | * | | | |
| 96 | 34 | | | | |
| 96 | 36 | | | | |
| 96 | 38 | * | | | |
| 96 | 40 | | | | |

**q = 3**

| n | m | | n | m | | n | m | |
|---|---|---|---|---|---|---|---|---|
| 12 | 6 | | 96 | 48 | | 400 | 48 | * |
| 12 | 12 | | 96 | 60 | | 400 | 60 | |
| 12 | 18 | | 144 | 30 | | 400 | 90 | |
| 20 | 12 | | 180 | 90 | | 400 | 192 | |
| 20 | 18 | | 192 | 6 | | 400 | 282 | |
| 20 | 30 | | 192 | 12 | | 400 | 384 | |
| 24 | 6 | | 192 | 18 | | 400 | 390 | |
| 24 | 12 | | 192 | 24 | | 720 | 90 | |
| 40 | 12 | | 192 | 30 | | 768 | 12 | |
| 40 | 18 | | 192 | 42 | | 768 | 24 | |
| 40 | 30 | | 192 | 48 | | 768 | 48 | * |
| 40 | 42 | | 192 | 60 | | 1440 | 90 | |
| 40 | 48 | | 192 | 90 | * | 1536 | 24 | |
| 40 | 60 | | 192 | 192 | | 1536 | 48 | |
| 44 | 12 | | 288 | 12 | | 2880 | 90 | |
| 44 | 18 | | 288 | 18 | | 3072 | 48 | |
| 44 | 30 | | 288 | 30 | | 5760 | 90 | |
| 44 | 42 | | 288 | 42 | | | | |
| 44 | 48 | | 288 | 48 | | | | |
| 44 | 60 | | 288 | 60 | | | | |
| 48 | 6 | | 288 | 90 | | | | |
| 48 | 12 | | 288 | 192 | | | | |
| 48 | 18 | | 288 | 282 | * | | | |
| 48 | 24 | * | 360 | 90 | | | | |
| 48 | 30 | | 384 | 6 | | | | |
| 48 | 42 | | 384 | 12 | | | | |
| 48 | 48 | | 384 | 18 | | | | |
| 48 | 60 | | 384 | 24 | | | | |
| 64 | 12 | | 384 | 30 | | | | |
| 64 | 18 | | 384 | 42 | | | | |
| 64 | 24 | | 384 | 48 | | | | |
| 64 | 30 | | 384 | 60 | | | | |
| 64 | 42 | | 384 | 90 | | | | |
| 64 | 48 | | 384 | 192 | | | | |
| 96 | 6 | | 384 | 282 | * | | | |
| 96 | 12 | | 400 | 12 | | | | |
| 96 | 18 | | 400 | 18 | | | | |
| 96 | 24 | | 400 | 30 | | | | |
| 96 | 30 | | 400 | 42 | | | | |
| 96 | 42 | | | | | | | |

HYPER was optimized across the following set of design parameters. Asterisks (*) denote the restricted set of parameters that are available for Reed-Solomon Kautz-Singleton (RS-KS) code-based designs (see the Supplementary Material).

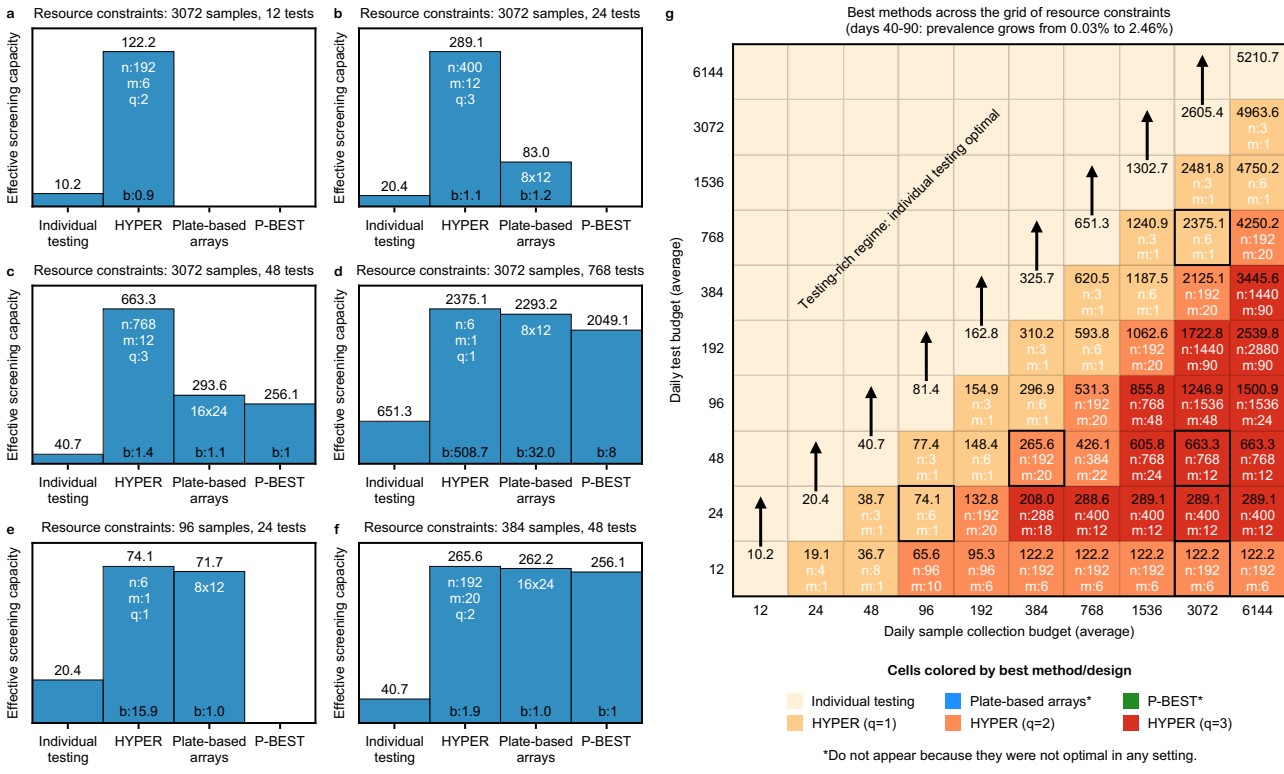

**Fig. 3 Comparison of pooling methods under resource constraints.** HYPER designs (Table 2) were evaluated together with individual testing, plate-based arrays[9], and P-BEST[8], across a range of sample collection and testing budgets. The basis for comparison was the effective screening capacity across days 40–90 of the simulation (Fig. 2), during which the prevalence increases exponentially from 0.03% to 2.46%. Bar plots on the left depict the effective screening capacities (bar height) in a testing-scarce setting (**a**), followed by increasingly testing-rich settings (**b**–**d**) and settings well-suited for the plate-based arrays and P-BEST (**e**, **f**). When multiple designs for a given method were available within the constraints (i.e., various choices of HYPER designs, or a choice between the 8 × 12 and 16 × 24 arrays), we use the most effective configuration and indicate it in white text within the appropriate bar. The average number of batches run per day is noted at the bottom of each bar. **g** Expanded comparison to a grid of sampling and testing budgets. Each cell is colored by the best method (where we separately identify HYPER designs with $q = 1, 2, 3$ splits in shades of orange/red), and shows the corresponding effective screening capacities (in black text). The best design configuration is written in white text. For HYPER, we write the number of individuals per batch $n$ and the number of pools $m$ for the best configuration; cell color already indicates the number of splits $q$. Note that $n$ and $m$ often do not match the daily sampling and testing budgets, respectively, since multiple batches can be run per day. The cases from (**a**–**f**) are outlined in black. See Supplementary Fig. 10 for additional details.

(Supplementary Fig. 12), or 1.36% (Supplementary Fig. 13), HYPER is consistently the most effective strategy across all settings. In an intermediate range with prevalence of 1.48% (Supplementary Fig. 14) and 2.46% (Supplementary Fig. 15) there is a subset of scenarios in which P-BEST outperforms HYPER, although the performance of each method is nearly equivalent in these settings. Outside these settings P-BEST is either not viable or substantially under-performs HYPER. At a higher prevalence of 3.15% (Supplementary Fig. 16) HYPER again performs best across all scenarios. We did not observe any scenarios in which plate-based arrays were most effective.

As before, since plate-based arrays[9] and P-BEST[8] are particular instances of general array-based and code-based designs, we also considered balanced arrays and RS-KS code-based designs (Supplementary Figs. 17 to 20). The balanced arrays were optimized over the same set of design parameters as HYPER (Table 2). On the other hand, RS-KS designs are available for only a limited subset (indicated in Table 2 by asterisks), so we also considered a Restricted HYPER that was optimized over the same subset. In our simulations, HYPER (and Restricted HYPER) were either more effective or about as effective as balanced arrays and RS-KS designs across the grid of resource constraints and were significantly more effective in important testing-constrained settings. Note also that the balanced arrays and RS-KS designs

we considered are actually extended variants that allow arbitrarily many individuals (see the Supplementary Material). HYPER outperforms the unextended variants even more since the most effective design parameters in testing-constrained settings often use many individuals. One might also consider forming variants of each design by concatenating $k$ disjoint copies to obtain a design with $kn$ individuals and $km$ pools. Indeed, such designs were already implicitly considered in the above analysis since they are equivalent to simply running $b = k$ batches.

## Discussion

In this paper, we present HYPER, a method for pooled testing with pooling designs based on hypergraph factorization. Our results demonstrate the effectiveness of this new family of pooling designs that are adaptable to any number of samples, with only mild conditions on the number of pools, while remaining maximally balanced in three senses (number of assignments per individual, pool, and combination of pools). This flexibility is critical to selecting appropriate designs under the widely varying global demands and capabilities for COVID-19 testing. In addition, the balanced nature of the designs ensures uniform treatment of samples and facilitates robust and simple implementation. Despite the simplicity of implementing HYPER, the existence and construction of the designs relies on deep mathematical results from combinatorics.

Our evaluation of HYPER in both the general statistical framework and the COVID-19 simulation can be used to guide the choice of design, depending on the setting and purpose of testing. For the general statistical setup, where each test has specificity $1 - \alpha$ and sensitivity $\beta$ independent of all other tests, we characterized the overall efficiency and accuracy of HYPER. Notably, we showed that HYPER has sensitivity $\beta^q$ independent of the pool sizes. Moreover, in the noiseless case, we showed that using roughly $m/n \approx 2p^{2/3} - p$ pools per individual maximizes the efficiency of HYPER designs. From our simulations, which model various realistic aspects of COVID-19 testing, we found a general trade-off between efficiency and sensitivity. Under this model, using larger pools often yields greater efficiency at low prevalence but also dilutes samples and results in lower sensitivity. One must consider how to balance these two aspects, and we discuss how optimizing the effective screening capacity captures both in a meaningful way.

Given the potential application of HYPER to future epidemics, it is important to also consider changes in test characteristics or epidemic dynamics. In our simulations (Supplementary Fig. 8), increasing the testing sensitivity (by reducing the limit of detection) led to an increase in sensitivity for all the methods, without much change in their relative performance. We also considered a change in epidemic dynamics that models a sustained two-wave epidemic[11]. In our simulations (Supplementary Fig. 9), the performance of all the methods varied from one phase of the epidemic to the next, but again without much change in their relative performance. Notably, sensitivity for all the methods (including individual testing) was lower during the decline phase than the two growth phases, even with matching prevalence. The viral loads of infected individuals were generally smaller (and hence harder to detect) during epidemic decline than they were during epidemic growth, due to a shift away from recent infections. Alternatively, if the viral kinetics change so that viral loads peak later, the smaller viral loads of recently infected individuals may lead to reduced sensitivity and may alter the difference between epidemic growth and decline. We expect that these changes would again affect all methods concordantly. Another important aspect is the rate of epidemic spread; prevalence for a slowly spreading epidemic remains low for a longer time, making it possible to use HYPER designs that sacrifice efficiency at high prevalence to dramatically increase efficiency at low prevalence. Overall, while a future epidemic would likely require some reevaluation to carefully account for its specific features, we expect the relative performance of all the methods to remain similar, making HYPER a promising candidate for future epidemics as well.

While pooled testing can substantially increase effectiveness depending on laboratory capacity and prevalence, it is important to also consider the added logistical challenges. Notably, the gains in testing effectiveness that we demonstrate above do not account for the additional pipetting steps during pooling, or the logistical cost of temporarily storing and retrieving samples for stage 2 testing. However, simple (Dorfman) pooling designs are receiving increasing interest[4–7,17,65] for real-world testing, demonstrating that these logistical challenges can be overcome in practice in a variety of settings. In comparison to Dorfman designs, more complex designs (with $q > 1$) will require up to $q$ times as many pipetting steps during stage 1 pooling. Depending on the relative timing and cost of each step in the protocol, this may shift the relative favorability of the strategies considered above. In particular, P-BEST, with $q = 6$ or more, may become relatively unfavorable if pooling steps are expensive, while plate-based arrays, which utilize multichannel pipettes, may become more favorable.

An important strength of the conservative decoder we used here is its conceptual simplicity, which can help reduce the risk of mistakes in practice. Moreover, it makes it possible to quickly illustrate (Fig. 1) and explain the method to those who may not yet be familiar with group testing. Notably, positives are only declared on the basis of a positive individual test, which can help make positive results easier to interpret. One can also consider using alternative decoders, e.g., that may offer more computationally efficient decoding or that may potentially reduce the number of stage 2 tests. For example, definite defective (DD) decoding[33,66] identifies putative positives like conservative decoding but then selects only those who are the only putative positive in a positive test. This decoder has the potential for higher efficiency since only the DD putative positives will then be tested in stage 2. However, in our analysis (Supplementary Fig. 21), doing so resulted in a significant loss of sensitivity, making the method less effective overall. Our analysis also considered using the DD decoder to instead only identify putative positives that can skip stage 2 and be declared positive. This approach preserved sensitivity but had a similar efficiency to conservative decoding so did not yield a significant improvement either. Exploring even more sophisticated decoders is an interesting direction for future work, though labs will need to assess whether the benefits outweigh the potentially greater complexity.

So far we have limited HYPER to $q \leq 3$. This has the advantage of reducing the additional logistical burden (and potential for error) that comes with splitting samples into more pools. Moreover, the efficient construction of hypergraph factorizations is highly nontrivial for $q > 3$. However, higher $q$ can have several advantages. For example, individuals in the same hyperedge (i.e., assigned to the same combination of pools) are identified as putative positives together as a block even if only one of them is actually positive. Using a higher $q$ can significantly increase the number of hyperedges $\binom{m}{q}$, reducing the number of individuals sharing a single hyperedge. Results for HYPER here also indicated that high $q$ designs can be highly effective when the sample collection resources significantly outstrip the testing resources, consistent with earlier studies of random assignment[11]. Likewise, greater efficiency can be obtained by using a multi-stage approach with more than two stages, which is also more logistically challenging. In practice, one must weigh these opportunities for greater effectiveness against the increased complexity. Such designs may be especially promising for labs with access to robotic pipetting platforms.

To conclude, we present a simple, efficient and flexible pooled testing strategy that can be easily tailored and implemented without specialized expertise or equipment. To further facilitate implementation, we provide an online tool available at http://hyper.covid19-analysis.org that makes it easy to generate and carry out designs for a broad range of settings.

## Methods

**Maximal balance, HYPER, and extensions.** As described in "Results", HYPER provides a simple and flexible pooled testing method that is maximally balanced. Supplementary Note S1 describes maximal balance in greater detail; it contains both examples (illustrating each of the three balance conditions) and a formal mathematical definition. Developing maximally balanced designs turns out to be nontrivial, as noted above in "Results". Supplementary Note S2 describes some of the challenges by considering various existing approaches. It discusses a couple straightforward but illustrative approaches (consecutive and lexicographic pooling), a couple randomized approaches (random assignment[11] and double-pooling[62]), exhaustive search, and finally, code-based and array-based approaches. Each approach falls short of adequately addressing this aspect of pooling design in some way. Supplementary Note S3 discusses how viewing this challenge through the lens of hypergraphs (as described above in "Results") leads naturally to an approach based on hypergraph factorization. Finally, Supplementary Note S4 describes the efficient constructions of hypergraph factorizations that we use in HYPER, Supplementary Note S5 describes how the HYPER design fits into the broader context of design theory, Supplementary Note S6 describes a more convenient way of presenting HYPER designs for implementation in the lab, and

Supplementary Note S7 describes how the decoder can be extended for error-correction of false negatives.

**Balanced arrays and Reed-Solomon Kautz-Singleton (RS-KS) code-based designs.** In addition to plate-based arrays[9] and P-BEST[8], we also considered general balanced arrays and Reed-Solomon Kautz-Singleton (RS-KS) code-based designs. The balanced arrays we considered use two-way ($q = 2$) and three-way ($q = 3$) arrays and make the pools balanced by using square/cube arrays and filling them in a carefully chosen order. See Supplementary Note S8 for more details and examples. The RS-KS code-based designs we considered form pools from Reed-Solomon[67] codes using the celebrated Kautz-Singleton[31] construction. See Supplementary Note S9 for more details and an example. As noted above in "Results", the balanced array and RS-KS code-based designs we considered are in fact extended variants that allow arbitrarily many individuals; see Supplementary Notes S8 and S9 for more discussion.

**Performance characterization under a common statistical model.** As presented above in "Results", we analyzed the performance of HYPER under the common statistical model where each individual (or in more general contexts, each sample) is positive independently at random with probability $p$ and each test may be incorrect with some probability, i.e., each test has a specificity of $1 - \alpha$ and a sensitivity of $\beta$. Supplementary Note S10 provides both the detailed derivations of these results (including a sharper bound on the expected number of tests used by HYPER) and further discussion of related works on optimality for group testing.

**Simulation under the COVID-19 model.** We performed simulations studies using the COVID-19 model of Cleary and Hay et al.[11]. The model first simulates viral loads for a large population of $n_{pop} = 12{,}500{,}000$ individuals across $d_{pop} = 357$ days during which the epidemic grows then declines. It captures the evolution of both: (a) viral loads within each individual, i.e., within-host viral kinetics, and (b) infection prevalence in the overall population. See Cleary and Hay et al.[11] for a detailed description. The main output we use is a matrix $Z^{(pop)} \in \mathbb{R}^{n_{pop} \times d_{pop}}$ of the population viral loads, where $z_{i,d}^{(pop)}$ is the viral load of individual $i$ on day $d$.

Next, the model simulates pooled testing to determine the average efficiency (relative to individual testing) and average sensitivity for each day. For the reader's benefit, we detail the process here. For HYPER designs, i.e., $H_{n,m,q}$, the simulation proceeds for each trial $r$ of day $d$ as follows:

1. Draw $n$ individuals uniformly at random from the population. Let $z_1, \ldots, z_n$ be their viral loads that day. That is, draw $n$ indices $k_1, \ldots, k_n$ uniformly at random from the set $\{1, \ldots, n_{pop}\}$ (with replacement), and let $z_i = z_{k_i,d}^{(pop)}$. Put another way, $z_1, \ldots, z_n \overset{iid}{\sim} \text{Uniform}(z_{1,d}^{(pop)}, \ldots, z_{n_{pop},d}^{(pop)})$. Individuals with nonzero viral load are positive/infected.

2. Generate the sampled viral load for each of the $m$ pools $\mathcal{I}_1, \ldots, \mathcal{I}_m \subseteq \{1, \ldots, n\}$ as follows:

$$v_j = \sum_{i \in \mathcal{I}_j} \text{Poisson}(z_i/|\mathcal{I}_j|), \qquad j = 1, \ldots, m,$$

where $|\mathcal{I}_j|$ is the size of pool $j$, i.e., the number of individuals assigned to it.

3. Compute stage 1 pooled testing results:
   - if $v_j > \text{LOD}$ then pool $j$ tests positive, where the LOD (limit of detection) we use is 100.
   - otherwise, pool $j$ tests negative with probability 0.99 (i.e., the false-positive rate of PCR results is 1%).

4. Select putative positives as those individuals that are not in any negative pools.

5. Compute stage 2 individual testing results for the putative positives: putative positive individual $j$ tests positive if $z_j > \text{LOD}$ and tests negative otherwise.

6. Declare individuals identified by HYPER as those that tested positive in stage 2.

7. Record the following for the current trial $r$ and day $d$:
   - the number of true positive individuals identified by HYPER: $n_{iden}^{(r)}(d)$,
   - the number of tests expended: $T^{(r)}(d) = m +$ number of tests used in stage 2,
   - the number of true positive individuals seen: $n_{pos}^{(r)}(d) =$ number of individuals with viral load $> 0$.

For each day, we repeat this for 500 initial trials, then continue until either at least 2500 true positive individuals have been seen or a total of 200,000 trials have elapsed (including the initial 500). This is to reduce experimental noise. Denoting $R$ to be the total number of trials run, we then compute the following averages across trials

$$\bar{T}(d) = \frac{1}{R}\sum_{r=1}^{R} T^{(r)}(d), \quad \bar{n}_{iden}(d) = \frac{1}{R}\sum_{r=1}^{R} n_{iden}^{(r)}(d), \quad \bar{n}_{pos}(d) = \frac{1}{R}\sum_{r=1}^{R} n_{pos}^{(r)}(d),$$

then finally compute the average efficiency (relative to individual testing) and

average sensitivity for day $d$ as follows:

$$\overline{\text{efficiency}}(d) = n/\bar{T}(d), \qquad \overline{\text{sensitivity}}(d) = \bar{n}_{iden}(d)/\bar{n}_{pos}(d).$$

Note that step 2 in the simulation above captures dilution due to pooling, since each individual's viral load gets divided by the pool size. The Poisson distribution models the arrival of viral particles when the small volume is pipetted from each swab. Note also that step 5 models the individual testing of stage 2 as having no false positives. Doing so simplifies the simulation without meaningfully affecting our conclusions (e.g., the most effective pooling designs, which do not depend substantially on stage 2 specificity). We do include false positives in stage 1, since the overall efficiency depends on the specificity there. The parameters were chosen to match earlier modeling studies[11,68–70].

For the $8 \times 12$ and $16 \times 24$ plate-based array designs[9], the simulation proceeds in the same way except for step 2, where the corresponding array pools are used instead. Recall that the array method is a two-stage method like HYPER. For P-BEST[8], which is a one-stage method, steps 1–3 are the same (except that step 2 now uses the P-BEST pools). Steps 4–6 are replaced by running the P-BEST decoder to identify individuals. For this, we followed the example (including its tuning parameters) provided online by the authors at https://github.com/NoamShental/PBEST/blob/f7ffebe6c7021ee40167239210806c5a1319f81e/mFiles/example_PBEST.m. Finally, since P-BEST has no second stage of validation tests, the number of tests expended is always $T^{(r)}(d) = m = 48$. Figure 2 plots the average efficiencies and average sensitivities of the various methods for each day in a 90-day window of epidemic growth. Here we included individual testing, which has a constant average efficiency of 1 (unity) since it is the baseline. Its average sensitivity on day $d$ is equal to

$$\overline{\text{sensitivity}}(d) = \frac{\text{Number of individuals (on that day) with viral load} > \text{LOD}}{\text{Number of individuals (on that day) with viral load} > 0},$$

since individual testing identifies those individuals with viral load $> \text{LOD}$, and true positive individuals are those with viral load $> 0$ (as before). The average sensitivities of the various methods appeared to generally have significant experimental noise. So, Figure 2 plots the raw averages (i.e., $\overline{\text{sensitivity}}(d)$) as dots along with a degree-8 polynomial curve fitted to $\overline{\text{sensitivity}}(d)$ vs. $\log_{10}p(d)$ across the plotting window of days $d = 20, \ldots, 110$, where $p(d)$ is the prevalence on day $d$.

In Fig. 2a, b, we compared HYPER designs $H_{96,16,2}$ and $H_{384,32,2}$ with their counterpart array designs and P-BEST. For the HYPER designs, the numbers $n$ of individuals per batch were chosen to match the array designs and P-BEST. The numbers $m$ of pools were chosen so that the corresponding pool sizes $nq/m$ match the maximum pool sizes of the array designs (12 for the $8 \times 12$ array and 24 for the $16 \times 24$ array). Figure 2c compares HYPER designs $H_{384,32,2}$, $H_{384,16,2}$, and $H_{384,12,2}$ that have varying numbers of pools. Figure 2d compares HYPER designs $H_{384,12,1}$, $H_{384,12,2}$, and $H_{384,12,3}$ that have varying numbers of splits.

**Comparison of pooling methods under resource constraints.** We used the simulations above to evaluate the various methods (individual testing, HYPER, plate-based array designs, P-BEST) under resource constraints and over time. We considered two forms of resource constraints: (i) a limited daily sample collection budget, and (ii) a limited daily testing budget. We let both range from 12 to 6144, forming the grid of resource-constrained scenarios shown in Fig. 3g and Supplementary Fig. 10, with a few selected scenarios highlighted in Fig. 3a to f. These figures evaluate average performance of the various methods when deployed across days 40–90 of the simulation. Supplementary Figs. 11 to 16 repeat the analysis (using the same set of scenarios) for individual days, namely days 53, 80, 83, 84, 90, and 93. Hence, we will focus on describing Fig. 3 and Supplementary Fig. 10; Supplementary Figs. 11 to 16 are similar.

In each scenario, we evaluated each method by its effective screening capacity $\bar{C}$ across a set of days $\mathcal{D}$. As discussed in the "Results", this performance metric measures how many individuals the method can screen under the resource constraints, with a correction applied to account for the associated sensitivity. Figure 3 and Supplementary Fig. 10 consider days 40–90, so $\mathcal{D} = \{40, \ldots, 90\}$ there. Supplementary Figs. 11 to 16 examine individual days, which corresponds, e.g., to $\mathcal{D} = \{53\}$ in Supplementary Fig. 11. To compute the effective screening capacity, we first determine the number of batches $b(d)$ that can be run on each day $d$, and its corresponding average $\bar{b}$:

$$b(d) = \min\left\{\frac{\text{sample collection budget}}{n}, \frac{\text{testing budget}}{\bar{T}(d)}\right\}, \qquad \bar{b} = \frac{1}{|\mathcal{D}|}\sum_{d \in \mathcal{D}} b(d).$$

If $\bar{b} < 0.9$ batches per day, i.e., fewer than 0.9 batches can be run per day on average, then the method is considered infeasible within the resource constraints and we set the method to have an effective screening capacity of $\bar{C} = 0$. Setting the above threshold at 0.9 captures an assumed flexibility to use fewer or more tests across days. Otherwise, if $\bar{b} \geq 0.9$, we compute the effective screening capacity $C(d)$ for each day $d$ and the effective screening capacity $\bar{C}$ across the days $\mathcal{D}$ as follows:

$$C(d) = \underbrace{n \times b(d)}_{\text{\#individuals screened}} \times \overline{\text{sensitivity}}(d), \qquad \bar{C} = \frac{1}{|\mathcal{D}|}\sum_{d \in \mathcal{D}} C(d).$$

Figure 3a to f shows the effective screening capacities for the considered methods as

bars, with the corresponding average number of batches noted at the bottom of each bar. Multiple configurations are available for both the array method (the $8 \times 12$ and $16 \times 24$ array designs) and HYPER (various choices of $n$, $m$, and $q$). For these methods, we select the most effective among all configurations, i.e., the configuration with the highest effective screening capacity $\bar{C}$. For HYPER, in particular, we optimized over the configurations listed in Table 2. The chosen configuration is noted at the top of each bar in Fig. 3a to f.

Supplementary Fig. 10 shows the bar graphs for the full range of resource-constrained scenarios considered. Figure 3g summarizes these findings by showing only which method was best (where we distinguish different choices of $q$ in HYPER), the corresponding effective screening capacity, and the corresponding configuration.

**Reporting summary**. Further information on research design is available in the Nature Research Reporting Summary linked to this article.

## Data availability

No raw data were collected in this study. The data and analyses generated in this study are available at https://github.com/dahong67/hyper-group-testing and can be regenerated using the accompanying code. The simulated population for the COVID-19 model was obtained from previously published code available at https://github.com/cleary-lab/covid19-group-tests.

## Code availability

Code is available on GitHub[71] (https://github.com/dahong67/hyper-group-testing).

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

## Acknowledgements

D.H. was supported in part by the Dean's Fund for Postdoctoral Research of the Wharton School, NSF BIGDATA grant IIS 1837992, and NSF Mathematical Sciences Postdoctoral Research Fellowship DMS 2103353. R.D. and X.L. were supported by a grant from the Partners in Health. E.D. was supported in part by NSF BIGDATA grant IIS 1837992.

## Author contributions

All authors contributed to the design of the study and to discussions of all aspects. D.H., R.D., and E.D. performed the theory development and analysis under the general statistical model. D.H., X.L., and B.C. performed and analyzed the simulations under the COVID-19 model. R.D. and X.L. developed the interactive online tool. D.H., B.C., and E.D. wrote the first draft of the manuscript. All authors reviewed and edited the manuscript.

## Competing interests

The authors declare no competing interests.
