## [Peer Review File · Nature Communications]

Reviewers' Comments:

Reviewer #1:

Remarks to the Author:

What are the major claims of the paper?

The authors present an explicit two-stage group testing procedure that outperforms some other known schemes. The suggested algorithms are good for a small amount of samples/tests which might be important for SARS-CoC-2 screening.

Are they novel and will they be of interest to others in the community and the wider field?

In my opinion, the results might be important for labs that test potentially infected people and don't have a really big power ($<10^3$ samples and $<10^2$ tests). I am not an expert in this field and don't know whether two-stage group testing can be easily applied in real lab testing. As for the information theory community, the results are less interesting and don't represent solid research.

If the conclusions are not original, it would be helpful if you could provide relevant references. Is the work convincing, and if not, what further evidence would be required to strengthen the conclusions?

The conclusions are original and the work is convincing. However, an additional revision is required in my opinion.

On a more subjective note, do you feel that the paper will influence thinking in the field?

I don't think that any breakthrough or novel idea was suggested in the paper. Nevertheless, I am glad to see that some researchers are more focused on practical schemes that can help in the real-world.

In general, the paper is written in a good way so that most of the ideas can be easily understood by the reader.

Many additional comments and suggestions are depicted below:

1) Some really important references are missing in the paper. In the following paper, it was suggested to use so-called d-disjunctive (superimposed) matrices for testing potentially infected people. This framework works without an error under the assumption that the total number of infected people is at most d among the population. In the provided paper, the best-known bounds on the size of d-disjunctive matrices were constructed. Additionally, classic error-correcting codes (such as Reed-Solomon codes) were first suggested to use for group testing. Later these constructions were shown to be order-optimal for several settings.

[1] Kautz, William, and Roy Singleton. "Nonrandom binary superimposed codes." IEEE Transactions on Information Theory 10, no. 4 (1964): 363-377.

[2] Erdős, Paul, Peter Frankl, and Zoltán Füredi. "Families of finite sets in which no set is covered by the union of r others." Israel Journal of Mathematics 51, no. 1-2 (1985): 79-89.

[3] Dyachkov, Arkadii Georgievich, and Vladimir Vasil'evich Rykov. "A survey of superimposed code theory." Problems of Control and Information Theory 12, no. 4 (1983): 1-13.

2) Array pools and q-dim. hypercubes: It is not clear from the writing why these approaches are potentially more efficient. Please highlight what makes it possible to have fewer tests or a larger probability of correct decoding. In addition, it is not clear how the hypercube approach is organized. As a potential reader, I would be glad to see a bit more details on this approach.

3) It is highlighted in the paper that your approach can be easily implemented and realized if a lab assistant has only a pencil and paper. Frankly speaking, most of the approaches (almost all) provide this feature as well. At the first stage, we just need to get rid of the samples that are not consistent with the test outcome. The complexity of this approach for your design is $O(q*n)$. This is not a phenomenal result since theoretically there exist some approaches with complexity $O(m)$. So, be honest about your methodology; do not oversell its potential.

- 4) The need of balanced designs. Please note that point iii) in your requirements automatically implies point ii) (in the asymptotic regime). I think that the need of balanced designs should be written in a more precise and clear way and some real statistics should confirm your conclusion. The arguments at the bottom of page 4 are not really rigorous and "worse efficiency" should be replaced by "worse WORST-CASE efficiency". On average, it still might be much better than a balanced design.
- 5) Discussions about random constructions. I don't understand a real drawback of theoretical limitations proved by the probabilistic arguments. You are writing that they are good with high probability. But, for a real application one needs to construct them only once and keep the results (e.g. like you suggest on the website). If a lab needs the testing scheme, then they can re-use a testing matrix stored or implemented somewhere. It is hard but still possible to work with balanced matrices when working with random arguments or implementing random designs. Whereas you consider only binary matrices with constant weight columns.
- 6) Terms specificity and sensitivity. These terms are not common in the group testing literature and appear in your paper on page 9 for the first time. I suggest describing the model you analyzed in more detail (page 9) and explain the meaning of these terms. I understand these terms only because I read the proofs.
- 7) Effective screening capacity. I actually don't understand why you concentrate on this quantity. I understand that any good metrics should reflect both the sensitivity and the number of individuals but the multiplication of these values are not necessarily the unique right choice. Also, it would be great if you emphasize what is the actual goal of your research (page 13). By fixing the number of tests and the probability of being infected, you want to maximize the effective screening capacity.
- 8) Baseline. In my opinion, it would be great if you add a fully adaptive scheme as a baseline. This strategy is definitely cannot be implemented in the real-life, but the readers can really realize how far your construction from the optimal strategy.
- 9) Parameters in simulations. I don't understand why the Poisson distribution is used in simulations (why not any other distribution with the same mean). Also, provide some references saying that your chosen parameters are close to real-life. Also, it is not clear why the simulation model is different from the one you are analyzing theoretically.
- 10) Block designs. I believe that many known block designs satisfying your requirements exist in the literature. From the writing, I am not convinced that you are the first who suggest such designs.

Supplementary materials:

- 1) What is the distribution of z_1, \dots, z_n ? The phrase "n individuals are taken uniformly at random from the population" is not clear
- 2) Theorem 1: What is q here? Can it be any integer number? "the number of tests required"...required for what?
- 3) proof of Theorem 1: I think that the terminology (samples and pools) is mixing in this proof and it is hard to read it. For example, T_i is the indicator that the i -th sample is positive (maybe i -th pool?). G_j stands for groups containing sample j (pools containing j -th item or items included to the j -th pool?). The index i in R_i is fixed, but I don't see any dependence of that in A_1, \dots, A_n . I recommend to proofread the proof, because I wasn't able to read it because don't see connection between formulas, notations and conclusions.
- 4) Please be accurate when saying that you have something optimal as it requires an additional clarification.

Typos:

- p. 17 "Designs with fewer pools have larger pools"
- Supplementary materials
- p.7 "a fixed-point-free map or order three"
- p.12 the specific rate .. can lead to better rates

Nikita Polianskii

Reviewer #2:

Remarks to the Author:

Use of pooled testing (or "group testing") as an alternative to individual testing when screening for a disease is a well established practice, going back to the work of Dorfman [17]. (Here and throughout, numbers in brackets are references in the paper, while letters in brackets are my own references, listed below.)

As a mathematical problem, there are various optimality or near-optimality results available: for one-stage testing [47, 48], two-stage testing [A], for multi-stage testing [B], and so on – the 2019 survey paper [30] is a good guide to these and many other recent results. However, these results are typically asymptotic as the number of individuals n tends to infinity, and based on random constructions. Less is known about what the best designs are for small finite n . A simple suggestion is the grid design (here referred to as "array pooling") – here the authors use non-square grids for comparison, for reasons that weren't clear to me. A generalisation of the array design is the so-called "hypercube" design, as used recently in an algorithm from Rwanda [8]. Another popular suggestion is designs based on error-correcting codes, an idea that goes back to the work of Kautz and Singleton in the 1960s [C]; one particular instance of this general framework is used by a group from Israel in an algorithm known as P-BEST [6].

This paper proposes some designs based on the combinatorial idea of hypergraph factorisation, and analyses them as part of a two-stage algorithm. My reading of the paper is that the authors are aiming for a sweet spot of "better than the simpler array design, and simpler than the complicated Kautz-Singleton designs".

The designs considered allow for each individual to be tested in $q = 1, 2,$ or 3 pools. For $q = 1$, there's nothing interesting to say beyond Dorfman's original work [17], so we concentrate on $q = 2, 3$ here. The mathematically optimal number of times to test an item is of the order $\log(1/p)$ for small prevalence p , so when the prevalence gets very low, larger q than 3 is mathematically preferable, although there may be practical reasons to prefer smaller q . (This is as good a point as any to note that your reviewer is a mathematician, and is not well placed to comment on practical issues within the laboratory that group testing procedures may have – I hope other reviewers can weigh in on those points.)

This paper seeks designs that are "balanced". It would have been helpful to have a clear formal mathematical "Definition 1: A pool design X is 'maximally balanced' if..." early on in the paper. I think that, when the numbers are such that an array design exists, ie $q = 2$ and n is (a multiple of) m^2 , then the array design is maximally balanced. Indeed, I think it's also true when n is between $m^2 - m$ and m^2 , using an "array with holes". But the paper doesn't seem to discuss this point. Is it correct that the $q = 2$ HYPER designs are an improvement on the array design due to them existing for all n , but are no better when the numbers allow an array design, or have I misunderstood?

I think the paper is proposing "trivial" two-stage testing, rather than "non-trivial" (or "conservative" rather than "non-conservative", in the terminology of [A]), although the example wasn't clear on this point. That is, if a positive test contains only one putative positive, does that putative positive still require an individual test, according to this algorithm? I think the answer is yes, but, having read the paper, I'm still unsure.

This paper is serious mathematics, and appears to be accurate throughout. The paper is well written, with a thorough (perhaps even too thorough!) literature review. However, I have two main points of concern. First, I don't feel the results give me a good enough like-with-like comparison to other existing methods. Second, I'm not sure Nature Communications is the best venue for this paper to appear when (as it surely should be) it is published.

To give an example of my first point, take Figure 2(a) This compares an $n = 96, m = 16, q = 2$ HYPER design to an array design. I have a number of questions: 1) Why was an unbalanced 8×10 array chosen for comparison over a balanced 10×10 array with 4 "holes"? 2) Why was it not compared with a random $n = 96, m = 16, q = 2$ design? 3) Why was it not compared with a random $n = 96, m = 16, q = 2$ design with the additional constraint that each pool contains 12 samples? 4) Why was it not compared with a code-based design with the same parameters $n = 96, m = 16, q = 2$? (Perhaps no such code based design exists? Then what about a truncated

code-based design?) Without such comparisons, how am I meant to know how good the HYPER design is? Similarly, in the top-right (unlabelled) graph, we get a comparison with another unbalanced array and one particular code-based design (P-BEST) with completely different parameters. Unsurprisingly, the design with completely different parameters has completely different behaviour, but I don't see that the reader learns much from that. Subfigures 2b and 2c don't have any comparisons at all with other designs. I don't think the paper can be accepted without such comparisons to existing designs in the literature. How else is the reader meant to judge if the HYPER designs are good or not?

My suspicion is that the HYPER designs will have extremely good performance, which is why I'm positive on the paper as a whole. But I'm not sure whether they will be "slightly-but-undeniably better than random designs" or "loads better than random designs", and similarly versus code-based designs. I wish the paper had given me the information to know.

While we're here, the "exponentially increasing prevalence over time" model on the x-axis for the Figure was a bit too clever-clever for my taste – a simple range of plausible prevalences on the x-axis would be fine, without trying to shoehorn it into some purported pandemic model.

Second, we have the question of whether Nature Communications is the best venue for this paper. This is a lengthy paper (23 pages – or 49 pages with full supplementary information) that is quite mathematically technical. It presents evidence from computer simulations, rather than actual laboratory studies. I would normally recommend such a paper goes to a combinatorial journal – something like "Combinatorics, Probability and Computing" or "Discrete Applied Probability" or "Journal of Combinatorial Optimization". Understandably (and admirably!) the authors wish to advertise their web app to practitioners, but I'm not sure a long technical paper in Nature Communications is the best way to advertise this useful tool.

If this paper had arrived in my inbox at one of those journals (and with appropriate comparisons with existing designs), I wouldn't hesitate to recommend acceptance. However, as we are, I narrowly come down on the side of rejection. If the editor feels - contrary to my thoughts above - that Nature Communications *is* in fact an appropriate venue for a paper such as this, I don't at all object, but would like some small revisions to see better comparisons with those existing designs before acceptance.

REFERENCES

- [A] Aldridge, Conservative two-stage group testing, arXiv
- [B] Zaman-Pippenger, Asymptotic analysis of optimal..., Prob Eng Inf Sci
- [C] Kautz-Singleton, Nonrandom binary superimposed codes, IEEE Trans Inf Th

Reviewer #3:

Remarks to the Author:

Dear Editor Dr Righetto,

Thank you for asking me to review the research article "HYPER: Group testing via hypergraph factorization applied to COVID-19" by Hong D et al (NCOMMS-21-07610-T). As instructed, my comments are focused on the utility of the method and its potential of use among clinical practitioners, rather than the technical details.

Pooled testing of samples may help to alleviate the heightened demand for large-scale testing during pandemics, which is especially challenging for resource-limited facilities / areas. In this study, the authors proposed a new group testing method (HYPER) – a two-stage pooling strategy based on hypergraph factorizations. Using simulation studies, they predicted that HYPER may outperform other methods especially in resource-constrained environments, and required little expertise for implementation.

Strengths:

Important and timely question as pooled testing is a valuable method to help alleviate the burden of laboratories especially in resource-limited areas during pandemics; particularly as the COVID-19 pandemic may linger on for the foreseeable future until adequate herd immunity is developed.

Major comments:

1. Results (Performance under a COVID-19 model): the authors focused the simulation during which the prevalence increases from 0.03% to 2.46% (days 40-90 in the simulation); and applied individual testing sensitivity of ~85%. This reviewer considers that additional simulation with the following parameters to be especially relevant for clinical use:

i) Evaluating the post-exponential phase; ie: a later phase in the simulation. As evident by the COVID-19 pandemic, it is often difficult to have coordinated laboratory testing in the initial phase of pandemics. How would this additional simulation change the performance of HYPER and would this affect the perceived impact on the current COVID-19 pandemic which is now past the initial phase?

ii) Increasing the individual testing sensitivity to $\geq 90\%$. The gold standard of qRT-PCR used for diagnosing COVID-19 and other emerging viruses usually has sensitivity much higher than 85%. Therefore, the authors should show the simulation data using a higher individual testing sensitivity as well as determine the highest individual testing sensitivity which would make HYPER less desirable than individual testing.

2. Under the COVID-19 simulation model, the authors took into account a) SARS-CoV-2 viral load kinetics in infected patient, b) dilution of viral load during pooling, and c) evolution of infection prevalence in a large population over time. As HYPER may be a useful tool not only for COVID-19, but also future pandemics, can the authors provide simulation to predict how these parameters would affect the practicality of HYPER? For example:

a) Viral load kinetics: SARS-CoV-2 viral load tends to peak during the early phase of disease, while SARS-CoV-1 viral load peaks at around day 10. How would this different viral shedding pattern affect the performance of HYPER?

b) Dilution of viral load during pooling: this would likely remain more or less constant.

c) Evolution of infection prevalence in a large population over time: respiratory viruses such as coronaviruses tend to spread much faster than other emerging viruses transmitted by alternative routes (eg: fecal-oral or arthropod-borne). Would the longer time taken for the increase in prevalence of these alternative agents diminish the usefulness of HYPER?

3. Website (<http://hyper.covid19-analysis.org>): the number of subjects per batch (n) is limited between 6 and 300. Can the maximum n of 300 be increased further as many resource-limited areas have large populations which far exceeds 300 per batch (eg: single-source outbreak clusters of epidemiologically-linked COVID-19 cases of >300 patients have been well reported in different settings).

Overview

We thank the Editor for inviting a revision of our paper, and the Reviewers for their questions, comments and suggestions, which have strengthened the paper. For your convenience, we have copied the reviews below (*in black italics*) with our responses provided point-by-point (*in blue*).

We also indicate changes in the manuscript *in blue*.

Key points in our revision include:

- **Substantially revised discussion of the need for balanced designs.** We explained how balance leads to a more consistent pooling process, which is important for real life implementation (**pg. 4**). We also added a **new analysis** that shows how balance can lead to more consistent performance, i.e., uniform sensitivity and efficiency across the individuals in the design (**new Supp Fig 6, new discussions on pgs. 5, 14-15**).
- **New comparison of HYPER with a bound on the maximum achievable efficiency** for the conservative two-stage testing problem that it tackles (**new Supp Fig 2**). We found that **HYPER is fairly close to optimal for this problem** in the noiseless linear prevalence setting. We also compared with a different problem that removes the real-life constraints we consider and so allows any group testing method (e.g., fully-adaptive methods). Better efficiency is achievable in this problem since it is much less constrained. We have added a **new discussion** of these results on **pg. 11**.
- **Expanded analyses under the COVID-19 model** that add more HYPER designs (**new Supp Figs 3-4, new discussion on pgs. 13-14**) as well as two random designs (random assignment and double-pooling) and a balanced square array design with holes (**new Supp Fig 5, new discussion on pg. 14**). Overall, we found that the designs with similar pool size had similar sensitivity while the designs with the same number of pools had similar efficiency at low prevalence.
- **New analyses for different test and epidemic characteristics**, specifically a 25-fold lower limit of detection, which increases sensitivity to ~95% (**new Supp Fig 7**), and a sustained two-wave epidemic (**new Supp Fig 8**). **New discussions** for them are on **pg. 15**. In both settings, the primary impact was to sensitivity, which generally changed concordantly across all methods. We also added a **new discussion** of different viral kinetics and epidemic dynamics (**pg. 20**).
- **New discussions and revisions that clarify** array designs and hypercube (**pg. 3**), conservative decoding (**pg. 8**) and its benefits (**pg. 21**), the common (non-COVID) statistical model and its importance (**pgs. 9-10**), effective screening capacity (**pg. 17**), maximal balance (**Supp Methods, pgs. S1-S2**), and the derivations (**Supp Methods, pgs. S11-S12**).

We hope we have addressed all the comments. We look forward to hearing from you.

Response to Reviewer 1

What are the major claims of the paper?

The authors present an explicit two-stage group testing procedure that outperforms some other known schemes. The suggested algorithms are good for a small amount of samples/tests which might be important for SARS-CoC-2 screening.

Are they novel and will they be of interest to others in the community and the wider field?

In my opinion, the results might be important for labs that test potentially infected people and don't have a really big power ($<10^3$ samples and $< 10^2$ tests). I am not an expert in this field and don't know whether two-stage group testing can be easily applied in real lab testing. As for the information theory community, the results are less interesting and don't represent solid research.

If the conclusions are not original, it would be helpful if you could provide relevant references. Is the work convincing, and if not, what further evidence would be required to strengthen the conclusions?

The conclusions are original and the work is convincing. However, an additional revision is required in my opinion.

On a more subjective note, do you feel that the paper will influence thinking in the field?

I don't think that any breakthrough or novel idea was suggested in the paper. Nevertheless, I am glad to see that some researchers are more focused on practical schemes that can help in the real-world.

In general, the paper is written in a good way so that most of the ideas can be easily understood by the reader.

We thank the Reviewer for their questions and suggestions, which have helped us to clarify and improve the paper. We appreciate that the Reviewer values the contributions of our work on helping improve COVID-19 testing capacity in resource constraint settings.

Many additional comments and suggestions are depicted below:

1) Some really important references are missing in the paper. In the following paper, it was suggested to use so-called d -disjunctive (superimposed) matrices for testing potentially infected people. This framework works without an error under the assumption that the total number of infected people is at most d among the population. In the provided paper, the best-known bounds on the size of d -disjunctive matrices were constructed. Additionally, classic

error-correcting codes (such as Reed-Solomon codes) were first suggested to use for group testing. Later these constructions were shown to be order-optimal for several settings.

[1] Kautz, William, and Roy Singleton. "Nonrandom binary superimposed codes." *IEEE Transactions on Information Theory* 10, no. 4 (1964): 363-377.

[2] Erdős, Paul, Peter Frankl, and Zoltán Füredi. "Families of finite sets in which no set is covered by the union of r others." *Israel Journal of Mathematics* 51, no. 1-2 (1985): 79-89.

[3] Dyachkov, Arkadii Georgievich, and Vladimir Vasil'evich Rykov. "A survey of superimposed code theory." *Problems of Control and Information Theory* 12, no. 4 (1983): 1-13.

We thank the Reviewer for these references and have included them in our introduction as references [31-33] on **pg. 2**, and added them to our expanded discussion of related results in the **Supplementary Methods (pg. S14)**.

2) *Array pools and q -dim. hypercubes: It is not clear from the writing why these approaches are potentially more efficient. Please highlight what makes it possible to have fewer tests or a larger probability of correct decoding. In addition, it is not clear how the hypercube approach is organized. As a potential reader, I would be glad to see a bit more details on this approach.*

Like other combinatorial methods, array pools and hypercubes place each individual in multiple pools. As a result, negative individuals in positive pools can be identified as long as any of their other pools test negative, thus avoiding the need to retest them. This makes it possible for these methods to use fewer tests than Dorfman testing, in which all individuals in a positive pool are putative positives that need to be retested in stage two.

We have revised our description to highlight this point in the **Introduction (pg. 3)**. We have also added some further clarifications on the hypercube approach.

3) *It is highlighted in the paper that your approach can be easily implemented and realized if a lab assistant has only a pencil and paper. Frankly speaking, most of the approaches (almost all) provide this feature as well. At the first stage, we just need to get rid of the samples that are not consistent with the test outcome. The complexity of this approach for your design is $O(q*n)$. This is not a phenomenal result since theoretically there exist some approaches with complexity $O(m)$. So, be honest about your methodology; do not oversell its potential.*

We agree with the Reviewer that our decoding approach is not the only one that is easily implemented. Indeed, the array design uses essentially the same decoder. However, some of the leading proposals in the context of COVID-19 testing do use more complicated decoders. For example, the decoder used in P-BEST involves sparse coding with tuned parameters. While it has the benefit of error-correction, it also comes at the cost of extra complexity (which may raise the barrier to entry). Hence, we felt that for practitioners considering these group testing methods, it was important to highlight that the decoding in HYPHER can indeed be done with pencil and paper or a simple spreadsheet.

We had not considered the computational complexity of the decoder, because we do not expect it to be a major issue in real life. Note that pipetting and running the tests accounts for much more of the overall time taken. Instead, we focused mainly on conceptual simplicity since this can be a major barrier to entry for those who may be new to group testing. Moreover, the conservative decoder we use has other important real-life benefits, such as only declaring a positive after a positive individual test. This can further help clinicians and patients interpret and act on positive results. We have clarified these motivations for conservative decoding in the **Discussion (pg. 21)**.

Nevertheless, the Reviewer's suggestion to be careful not to oversell is well taken. Indeed, theoretically there may exist approaches with better computational complexity, and we have made sure to note this in the paper (**pg. 21**). We have also added a note that other decoders with similar features to conservative decoding may exist and could also be of interest (**pg. 21**).

4) The need of balanced designs. Please note that point iii) in your requirements automatically implies point ii) (in the asymptotic regime).

If we understand correctly, the statement is as follows: for sufficiently large n , if a design satisfies requirement (iii), i.e., has maximally balanced pool combinations, then it also satisfies requirement (ii), i.e., has maximally balanced pools.

We do not see why this is true. Consider lexicographic designs with $q=2$. These designs assign individuals to pool pairs by cycling through all pairs in lexicographic order (i.e., AB, AC, AD, ...). By construction, these designs always have maximally balanced pool combinations. However, it seems that they have maximally balanced pools only if $n = -1, 0$ or $+1$ modulo $(m \text{ choose } 2)$.

As a concrete example, consider $m=4$ pools with $q=2$ splits. Then individuals are assigned as follows:

Individual	1	2	3	4	5	6	7	8	...
Assignment	AB	AC	AD	BC	BD	CD	AB	AC	...

As n grows, the corresponding pool sizes for this design are:

n	1	2	3	4	5	6	7	8	...
# in pool A	1	2	3	3	3	3	4	5	...
# in pool B	1	1	1	2	3	3	4	4	...
# in pool C	0	1	1	2	2	3	3	4	...
# in pool D	0	0	1	1	2	3	3	3	...

Only the highlighted values of n satisfy the condition of maximally balanced pools. It is not the case that it holds for sufficiently large n , and not necessarily even if n and m both go to infinity.

I think that the need of balanced designs should be written in a more precise and clear way and some real statistics should confirm your conclusion. The arguments at the bottom of page 4 are not really rigorous and "worse efficiency" should be replaced by "worse WORST-CASE efficiency". On average, it still might be much better than a balanced design.

We thank the Reviewer for this suggestion. We have reworked the explanation of the need for balanced designs to make it more precise and clear (**pgs. 4-5**).

Specifically, the need for balanced designs is a real-life need coming from our application to COVID-19 testing. For instance, having balanced pools means that the amount of volume to pipette for each sample is the same across pools. This simplifies the pipetting procedure and allows for more robust implementation and quality control.

Moreover, an important criterion in real life is that the performance be independent of where the positive individual happened to fall. This can fail without balanced designs, as explained below.

Uneven sensitivity is undesirable in real life because it means that positive individuals are more likely to be identified if they happen to "get lucky" and are placed in a more sensitive position in the design. In the context of testing for COVID-19, it is important that the design treats all individuals as uniformly as possible. Labs may feel uncomfortable with a procedure that dilutes samples unevenly and has better sensitivity for some individuals than others. In such cases, they may consider the worst sensitivity across all individuals to be most relevant in their decision-making. Note that the sensitivity loss due to dilution is a major source of concern for labs considering group testing. Uniform sensitivity means that individuals have the same sensitivity regardless of where they happened to be placed in the design.

Uneven efficiency is undesirable in real life because it means that the stage-two workload can vary depending on where positive individuals happen to fall in the design. This can make the logistics of testing more challenging for labs. Note that labs must plan and reserve enough tests for stage two. If the location of positive individuals (which is unknown a priori) affects the number of stage two tests, they may need to effectively plan for the worst case. However, doing so limits how many tests they can use in stage one since they must be conservative.

We have added a major **new simulation** investigating how different forms of imbalance affect efficiency and sensitivity (**Supp Fig 6, condensed discussion on pgs. 14-15**). It considers ($n=96$, $m=16$, $q=2$) designs where a single positive individual is placed in each of the n locations of the design. The positive viral load is drawn from the distribution of nonzero viral loads from day 80 (prevalence roughly 1.06%).

We compared the following designs:

- HYPER pooling, which has maximally balanced pools and pool combinations,

- Consecutive pooling (AB, CD, EF, ...), which has maximally balanced pools but not pool combinations (it only uses a subset of pool combinations),
- Lexicographic pooling (AB, AC, AD, ...), which has maximally balanced pool combinations but not pools,
- Random pooling (each individual assigned to q of the m pools selected uniformly at random), which guarantees neither maximally balanced pools nor pool combinations,
- Double-pooling (individuals randomly partitioned into m/q pools $q=2$ times), which guarantees maximally balanced pools but not pool combinations.

Since random pooling and double-pooling are both random designs, we consider three random draws for each, and report both the mean behavior and variability below.

Based on the simulation results, we found that:

- Consecutive pooling, which only uses a subset of pool combinations, was generally less efficient. It does, however, have balanced pools yielding uniform sensitivity regardless of where the positive individual fell.
- Lexicographic pooling has balanced pool combinations yielding uniformly high efficiency. However, it has unbalanced pools resulting in uneven sensitivity. Notably, a positive individual in location 93 had a sensitivity of 74.7%, while a positive individual in location 4 had a sensitivity of 73.3%.
- Random pooling had varying performance depending on which particular design gets drawn, and often had unbalanced pool combinations and pools. As a result, all three draws had uneven efficiency and sensitivity. For example, for draw 1, a positive individual in location 11 had a sensitivity of 75.7% while a positive individual in location 7 had a sensitivity of 72.2%.
- Double-pooling guarantees balanced pools but often has unbalanced pool combinations. As a result, all three draws had uniform sensitivity but uneven efficiency.
- HYPER had both uniformly high efficiency and uniform sensitivity. Moreover, its median efficiency (5.68 individuals/test) and median sensitivity (74.4%) were generally among the best, suggesting that it is not only better in the worst-case but also overall.

Overall, balancing the pool combinations was important for efficiency. This is because all samples with the same pool combination get retested when one of them is positive (and caught in stage one). Among the above designs, only HYPER and lexicographic designs guarantee this form of balance. For sensitivity, balancing the pool sizes is important. This is because dilution is a driving factor for sensitivity. Among the above designs, only HYPER, consecutive and double-pooling guarantee this form of balance. HYPER is the only design that guaranteed both forms of balance and had both uniform efficiency and sensitivity. Notably, this did not appear to come at the expense of overall efficiency and sensitivity. Indeed, HYPER was among the best overall for this experiment.

We remark that balance is important even beyond the present COVID setting, as reflected for instance in the other statistical model considered in our work. Note that under that model, efficiency would also vary depending on where the positive individual happened to fall for

imbalanced designs. However, sensitivity would not vary since sensitivity under that model does not depend on dilution.

5) Discussions about random constructions. I don't understand a real drawback of theoretical limitations proved by the probabilistic arguments. You are writing that they are good with high probability.

We think this was probably a typo, but we want to clarify that the probabilistic arguments show that the random construction is not maximally balanced, i.e., **not** good, with high probability.

An immediate drawback of not being maximally balanced is that the real-life benefits of consistency in the pooling process are lost. Volumes to be pipetted become varied, making the process more complicated and error-prone. As referenced above, we have reworked the discussion in the paper (**pg. 4**) to explain this more clearly. As illustrated in the new simulation (**Supp Fig 6**), random draws can also thus have uneven efficiency and sensitivity depending on where a positive individual happens to fall in the design. One could take many random draws to improve the chance of getting a maximally balanced design, but this does not guarantee a maximally balanced design. In contrast, HYPER directly gives a guaranteed maximally balanced design.

If we were unable to reliably produce maximally balanced designs, we could attempt to find something “good enough”. However, it is a bit unclear how close is close enough. It is not even obvious how to measure imbalance. HYPER obviates all these considerations since it simply produces maximally balanced designs from the start.

But, for a real application one needs to construct them only once and keep the results (e.g. like you suggest on the website). If a lab needs the testing scheme, then they can re-use a testing matrix stored or implemented somewhere.

The design needs of different labs can be quite diverse and may even change over time, e.g., when the prevalence grows or shrinks. Having the ability to easily choose different n, m, q enables them to better tailor the design to their situation and needs. One could indeed attempt to construct them once (e.g., using a randomized or exhaustive search) and keep the results, but this would then require anticipating all the n, m, q that could be needed. To give flexibility, one would then need to consider an incredibly large set of values. Our website <http://hyper.covid19-analysis.org> allows researchers to conveniently calculate the HYPER design for any given n, m, q . We have added a remark to clarify this in the paper (**pg. 6**).

For reference, even though our web tool limits users to m up to 48 and n up to 6144, this already corresponds to over 140 thousand designs for $q=2$ and over 42 thousand designs for $q=3$. With HYPER, it is simple for us to provide this broad range. Indeed, even broader ranges can be easily done. This is because we can compute the designs on-the-fly. If we were to instead try to precompute these, we would need to rerun the search for each choice of parameters. If the search was unsuccessful in finding a maximally balanced design, which may

be likely for large designs, we might then resort to hand-tweaking the design or perhaps simply accept something not maximally balanced. Designs that are “somewhat” but not maximally balanced could still be “good enough”, and quantifying deviation from imbalance and its exact impacts would be interesting to understand from a theoretical perspective. However, we note that the real-life need is somewhat obviated by HYPER since it provides a simple way to directly generate maximally balanced designs (for the range covered by our paper).

We also note that one could attempt to generate maximally balanced designs by tweaking other designs, e.g., partially filling larger designs. However, such approaches are somewhat ad-hoc and HYPER instead systematically generates maximally balanced designs.

It is hard but still possible to work with balanced matrices when working with random arguments or implementing random designs. Whereas you consider only binary matrices with constant weight columns.

Indeed, the double-pooling design discussed above is an example of a simple random design that is more balanced than the “random constant weight” design we focused on. It randomly partitions the individuals into pools twice, thus guaranteeing balanced pool sizes. However, as discussed above, it does not necessarily produce balanced pool combinations. We have added a remark to the paper to point this out (**pg. 6**).

We are unaware of a random design that produces maximal balance in all three senses (uniform splits, maximally balanced pools, maximally balanced pool combinations), and agree with the Reviewer that it is likely possible, but may be hard. Given this difficulty, we think HYPER is an attractive option as it provides a relatively simple way to directly obtain maximally balanced designs.

6) Terms specificity and sensitivity. These terms are not common in the group testing literature and appear in your paper on page 9 for the first time. I suggest describing the model you analyzed in more detail (page 9) and explain the meaning of these terms. I understand these terms only because I read the proofs.

Thanks for the suggestion. We have used them in the customary way, but agree that it is helpful to clarify them. We have described the model in more detail and defined the terms specificity and sensitivity (**pg. 9**).

7) Effective screening capacity. I actually don't understand why you concentrate on this quantity. I understand that any good metrics should reflect both the sensitivity and the number of individuals but the multiplication of these values are not necessarily the unique right choice.

Indeed, a good metric should reflect both sensitivity and the number of individuals screened. There are certainly other metrics that do so. However, a unique property of the effective screening capacity, i.e., the product of sensitivity and number of individuals screened, is that

multiplying it by prevalence yields the number of positive individuals identified by the screen. To see why, recall that

$$\begin{aligned} \text{sensitivity} &= \# \text{ positive individuals declared positive} / \# \text{ positive individuals screened} \\ \text{prevalence} &= \# \text{ positive individuals screened} / \# \text{ total individuals screened} \end{aligned}$$

So that the product of effective screening capacity and prevalence is

$$\begin{aligned} &\text{effective screening capacity} * \text{prevalence} \\ &= \text{sensitivity} * \# \text{ total individuals screened} * \text{prevalence} \\ &= (\# \text{ positive individuals declared positive} / \# \text{ positive individuals screened}) * (\# \text{ total} \\ &\text{individuals screened}) * (\# \text{ positive individuals screened} / \# \text{ total individuals screened}) \\ &= \# \text{ positive individuals declared positive} \end{aligned}$$

This makes it especially meaningful and useful in public health contexts, where the goal may be to find and isolate as many infected individuals as possible. We have added some of these clarifications to the paper (pg. 17).

Also, it would be great if you emphasize what is the actual goal of your research (page 13). By fixing the number of tests and the probability of being infected, you want to maximize the effective screening capacity.

To slightly clarify, the goal in this section is indeed to maximize the effective screening capacity, but it is not for a fixed number of tests alone or for a fixed probability of being infected (i.e., the prevalence). Instead we consider maximizing the effective screening capacity over a **window of days** during which not only does the prevalence change but also the distribution of nonzero viral loads. The constraint is for a fixed budget of how many tests can be performed (on average) and a fixed budget of how many samples can be collected. Namely, we seek to maximize the effective screening capacity over windows of time given resource constraints of limited amounts of sample collection and testing, as this closely corresponds to real-world decision making. We have clarified and emphasized this as suggested (pg. 17).

8) Baseline. In my opinion, it would be great if you add a fully adaptive scheme as a baseline. This strategy is definitely cannot be implemented in the real-life, but the readers can really realize how far your construction from the optimal strategy.

We thank the Reviewer for this interesting question. We share the desire to help readers understand what might be achievable and how close or far current methods are to it. However, we think that fully-adaptive schemes do not actually provide a meaningful baseline for HYPHER, because **fully-adaptive schemes solve a different problem from HYPHER**. Importantly, the problem solved by fully-adaptive schemes lacks real-life constraints that are central to the present paper (as the Reviewer notes). Thus, a more meaningful comparison is of the two problems tackled by fully-adaptive schemes and HYPHER, respectively. To this end, we have added a **new comparison** that: a) compares the two problems to help readers understand how

different they can be, b) illustrates that fully-adaptive schemes are near-optimal for the problem they tackle (as is well known), and c) assesses how close or far HYPER is to optimal for the problem (with real-life constraints) that it tackles.

In more detail, the two problems are:

- Problem 1: Devise an efficient group testing method (any # stages, any decoder, etc.). This problem is tackled by fully-adaptive methods.
- Problem 2: Devise an efficient two-stage method (implementable in real life, small number of splits) with a conservative decoder. This problem is tackled by HYPER and array methods.

As the Reviewer notes, solutions to problem 1 may not be implementable in real life for applications like COVID-19 testing. Multistage methods in particular can take too long to return the final results. Getting results back quickly is crucial when an epidemic is spreading, since it helps enable strategies for slowing the spread of infection. Hence, we instead tackled problem 2 with the goal (and hope!) of providing a method that can be used in real life.

As noted above, since the problems are different, we think comparing HYPER with fully-adaptive methods is not apples-to-apples and could be somewhat misleading. We think the more relevant question is how far or close HYPER is from the best possible efficiency for problem 2.

To this end, we have added a new figure (**Supp Fig 2**) that compares known lower bounds on the number of tests for the two problems, i.e., fundamental limits for both problems. Note that this is under the noiseless model where such results are available (also where we expect fully-adaptive methods to be efficient). We also consider the linear prevalence regime. We think it is more meaningful for settings like COVID-19 testing where a fixed proportion of individuals are infected and the proportion does not reduce as more individuals are sampled.

As one would naturally expect, the lower bound (counting bound, [36]) for problem 1 is lower than the lower bound [37] for problem 2 since it is much less constrained. For problem 1, fully-adaptive methods are nearly optimal, as was already known (e.g., [36]).

For problem 2, HYPER appears to in fact be somewhat close to optimal. We think the remaining gap visible for smaller prevalences is likely due to: a) looseness in our upper bound for HYPER, b) potential looseness in the lower bound [37], and c) our added constraint that $q \leq 3$ (to ease real-life implementation).

Further sharpening our analysis of HYPER (especially for $q=3$) is an interesting direction for future work, and may show that HYPER is in fact even closer to optimal than our current results show.

We thank the Reviewer for their interesting question that prompted us to consider the optimality of HYPER more carefully, and led us to discover that it is in fact closer to optimal (for the

relevant problem) than we realized. In addition to the new **Supp Fig 2**, we have added a discussion of this to the paper (**pg. 11**).

9) *Parameters in simulations. I don't understand why the Poisson distribution is used in simulations (why not any other distribution with the same mean).*

We use the Poisson distribution since it models the “arrival” of viral particles during the pipetting process. We have clarified this in the **Supplementary Methods (pg. S3)**.

Also, provide some references saying that your chosen parameters are close to real-life.

We have added references to the **Supplementary Methods (pg. S3)** pointing to relevant modeling studies [11;73-75].

Also, it is not clear why the simulation model is different from the one you are analyzing theoretically.

The COVID-19 model and “theoretical” model are different since they are assessing different aspects of HYPER. The “theoretical” model helped us to study the properties of HYPER under a common and general statistical setting, and importantly allowed us to compare with results for other methods (that are often analyzed under this model). The COVID-19 model helped us understand properties we could expect to see for COVID-19 testing, by incorporating various elements missing from the common statistical model in the literature. Importantly, it includes the effect of dilution, and the evolving distribution of viral loads over time. We felt that both were important to consider.

Note that we did also carry out simulations under the “theoretical” model (**Supp Fig 1**). A theoretical analysis under the COVID-19 model is a challenging future work due to some of the complexity of the model.

We have adjusted the section headings to clarify that the difference is in the models being considered. We have also added some discussion to explain that both are important to consider given our focus on the important problem of COVID-19 testing in addition to the more general applicability of HYPER (**pg. 9**).

10) *Block designs. I believe that many known block designs satisfying your requirements exist in the literature. From the writing, I am not convinced that you are the first who suggest such designs.*

We have looked and are unaware of any equally simple and flexible designs that are also maximally balanced. Balanced incomplete block designs are perhaps closest, as they are essentially equivalent for $q=2$, but this correspondence appears to fail for general q . If you are aware of some designs satisfying our requirements, we would be glad to learn about them!

We also do not claim to be the first to apply hypergraph factorization in the general context of group testing. We are aware of at least one paper (D'yachkov, Vorobyev, Polyanskii, Shchukin) that also uses hypergraph factorization but (as far as we understand) they use it in a different way. While we form a hypergraph where the pools are the vertices and hyperedges are pool assignments, their technique appears to form a hypergraph where individuals are the vertices and coloring of the graph is used to determine the tests to conduct in the second (of four) stages. We have added a remark clarifying the relationship to the paper (**pg. 8**).

Supplementary materials:

1) *What is the distribution of z_1, \dots, z_n ? The phrase "n individuals are taken uniformly at random from the population" is not clear*

We have clarified this in the **Supplementary Methods (pg. S2)**. On each day, we have a viral load z_{pop_i} for each individual i in the simulation population of $n_{pop}=12,500,000$ individuals. The z_1, \dots, z_n are i.i.d. draws of the uniform distribution on $(z_{pop_1}, \dots, z_{pop_{n_{pop}}})$.

2) *Theorem 1: What is q here? Can it be any integer number? "the number of tests required"...required for what?*

Indeed, q is any integer number here and denotes the number of pools assigned to each individual. The "number of tests required" refers to the number of tests that are used by HYPER (including the tests used in the second stage). We have rephrased it as "number of tests used" to clarify (**pg. S11**).

3) *proof of Theorem 1: I think that the terminology (samples and pools) is mixing in this proof and it is hard to read it. For example, T_i is the indicator that the i -th sample is positive (maybe i -th pool?). G_j stands for groups containing sample j (pools containing j -th item or items included to the j -th pool?). The index i in R_i is fixed, but I don't see any dependence of that in A_1, \dots, A_n . I recommend to proofread the proof, because I wasn't able to read it because don't see connection between formulas, notations and conclusions.*

We apologize for the notational issues in the proof. We went through the proof carefully, and cleared up some terminology (**pgs. S11-S12**):

- T_i indeed refers to samples
- G_j is the subset of all samples in pool j (here our original terminology "groups containing sample j " is suboptimal/incorrect, and your correction is right)
- For the notation, indeed A_j do not depend on i , but this is just for notational simplicity, and we have now explicitly mentioned this.

4) *Please be accurate when saying that you have something optimal as it requires an additional clarification.*

As suggested, we have gone through the paper and added clarifications. Please let us know if there are other places where we could make this more clear.

Note that, as discussed above, HYPER does appear to be somewhat close to optimal in the noiseless linear regime, where prevalence p is fixed as n goes to infinity (**Supp Fig 2**). That said, it does not appear to be completely optimal and closing this gap is future work.

Typos:

p. 17 "Designs with fewer pools have larger pools"

We think this was not a typo but was perhaps a bit confusing. We have rephrased it to say "Reducing the number of pools m means each pool will contain more individuals, leading to more dilution and lower sensitivity" (**pg. 19**).

Supplementary materials

p.7 "a fixed-point-free map of order three"

Corrected (**Supplementary Methods, pg. S8**). Now says "a fixed-point-free map of order three".

p.12 the specific rate .. can lead to better rates

We think this was not a typo but was perhaps a bit confusing. We have rephrased it to say "However, the best efficiency (and the algorithms that achieve it) depends on the specific rate at which $p \rightarrow 0$ " (**Supplementary Methods, pg. S14**).

Response to Reviewer 2

Use of pooled testing (or "group testing") as an alternative to individual testing when screening for a disease is a well established practice, going back to the work of Dorfman [17]. (Here and throughout, numbers in brackets are references in the paper, while letters in brackets are my own references, listed below.)

As a mathematical problem, there are various optimality or near-optimality results available: for one-stage testing [47, 48], two-stage testing [A], for multi-stage testing [B], and so on – the 2019 survey paper [30] is a good guide to these and many other recent results. However, these results are typically asymptotic as the number of individuals n tends to infinity, and based on random constructions. Less is known about what the best designs are for small finite n . A simple suggestion is the grid design (here referred to as "array pooling") – here the authors use non-square grids for comparison, for reasons that weren't clear to me. A generalisation of the array design is the so-called "hypercube" design, as used recently in an algorithm from Rwanda [8]. Another popular suggestion is designs based on error-correcting codes, an idea that goes back to the work of Kautz and Singleton in the 1960s [C]; one particular instance of this general framework is used by a group from Israel in an algorithm known as P-BEST [6].

This paper proposes some designs based on the combinatorial idea of hypergraph factorisation, and analyses them as part of a two-stage algorithm. My reading of the paper is that the authors are aiming for a sweet spot of "better than the simpler array design, and simpler than the complicated Kautz-Singleton designs".

The designs considered allow for each individual to be tested in $q = 1, 2,$ or 3 pools. For $q = 1$, there's nothing interesting to say beyond Dorfman's original work [17], so we concentrate on $q = 2, 3$ here. The mathematically optimal number of times to test an item is of the order $\log(1/p)$ for small prevalence p , so when the prevalence gets very low, larger q than 3 is mathematically preferable, although there may be practical reasons to prefer smaller q . (This is as good a point as any to note that your reviewer is a mathematician, and is not well placed to comment on practical issues within the laboratory that group testing procedures may have – I hope other reviewers can weigh in on those points.)

This paper seeks designs that are "balanced". It would have been helpful to have a clear formal mathematical "Definition 1: A pool design X is 'maximally balanced' if..." early on in the paper.

*We thank the Reviewer for this suggestion. We have added a formal mathematical definition (**Definition 1**) along with some illustrative examples to the **Supp Methods (pg. S1)**. We have also added a pointer to this formal definition early on in the paper (**pg. 4**).*

I think that, when the numbers are such that an array design exists, ie $q = 2$ and n is (a multiple of) m^2 , then the array design is maximally balanced.

We think the Reviewer may have meant that “n is (a multiple of) $(m/2)^2$ ”; note that a square array of m pools total has $m/2$ row pools and $m/2$ column pools. In case this interpretation is wrong, however, we will first address the claim as written then address the alternative below.

To the best of our understanding, the claim as written is not quite correct. For example, consider a 2x2 array. This has $m=4$ pools (2 rows + 2 columns) so we take $n=m^2=16$ individuals. This is more than the number of cells in the array, so we presume the Reviewer means to place 4 individuals in each of the cells.

For example, something like this:

1,2,3,4	5,6,7,8
9,10,11,12	13,14,15,16

Namely, individuals 1-4 are in the “row 1, column 1” cell, and so on. Labeling the row pools A and B and labeling the column pools C and D, this produces the following pool assignments:

Individual	1	2	3	4	5	6	7	8
Pools	AC	AC	AC	AC	AD	AD	AD	AD
Individual	9	10	11	12	13	14	15	16
Pools	BC	BC	BC	BC	BD	BD	BD	BD

The pools are indeed balanced since each pool contains 8 individuals. The pool combinations may also appear balanced at first because the pool combinations that are used (AC, AD, BC, BD) are all used four times. However, the pool combinations are not balanced overall; e.g., AB was never used while AC was used four times.

Next, we suppose the Reviewer meant “n is (a multiple of) $(m/2)^2$ ”. In this case, the statement is indeed correct when $n=(m/2)^2$ since all pool pairs are used either once or never. However, it is not true when n is a multiple of $(m/2)^2$ for the same reasons as in the 2x2 example above.

Indeed, I think it's also true when n is between $m^2 - m$ and m^2 , using an "array with holes". But the paper doesn't seem to discuss this point.

We presume again that m here is referring to $m/2$, the side length of the array. In this case, we agree that an array with holes can be maximally balanced for n between $(m/2)^2-(m/2)$.

As may already be apparent to the Reviewer, the holes must be placed carefully. For example, placing all the holes in the first row of the array will produce imbalanced pool sizes. One

approach is to place the holes along the main diagonal. We have added some discussion of this to the **Supplementary Methods (pg. S2)**.

Is it correct that the $q = 2$ HYPER designs are an improvement on the array design due to them existing for all n , but are no better when the numbers allow an array design, or have I misunderstood?

We think this is not quite the case. We have added a **new simulation (Supp Fig 3)** that adds HYPER designs with matching numbers of pools ($m=20$ for the 8×12 array and $m=40$ for the 16×24 array). In this case, the numbers ($n=96, m=20, q=2$ in the left panel and $n=384, m=40, q=2$ in the right panel) allow an array design. The added HYPER designs have nearly the same efficiency as the array designs. However, they have slightly higher sensitivity than their array counterparts. We have added some discussion of this to the paper (**pg. 13**).

That said, we do think that square versions of the array designs would perform fairly similarly to their HYPER counterparts. The lower sensitivity of the array designs is likely due to their imbalanced pool sizes. Hence, we do think that when the numbers allow for a square array, the performance will likely be similar to HYPER. That is to say, the corresponding $q=2$ HYPER designs will be no better but also no worse. As such, we do not see a performance benefit to using array designs even in these cases. Moreover, as we note below, a key strength of array designs is that they cohere with the plate sizes (8×12 and 16×24) and multi-channel pipettes that are common in labs. This is an important real-life benefit since it simplifies the pipetting for the pooling stage, but is largely lost if the standard plate sizes are not used.

I think the paper is proposing "trivial" two-stage testing, rather than "non-trivial" (or "conservative" rather than "non-conservative", in the terminology of [A]), although the example wasn't clear on this point. That is, if a positive test contains only one putative positive, does that putative positive still require an individual test, according to this algorithm? I think the answer is yes, but, having read the paper, I'm still unsure.

Yes, that is correct. We have clarified this in the paper (**pg. 8**).

We focused here on using a conservative decoder for its simplicity and robustness.

That said, one could also use the HYPER pooling design with more sophisticated decoders that identify definite positives, e.g., when a positive test contains only one putative positive. While this can improve efficiency, it also involves extra work and introduces opportunities for mistakes. Moreover, using more sophisticated decoders can make the method more difficult to illustrate and explain to those who may not yet be familiar with group testing. They may include labs trying to implement group testing for the first time (our target audience), as well as patients receiving the results. Hence, our focus on the conservative decoder.

Labs will need to weigh the potential benefits against the potential challenges in deciding what decoder is most appropriate for their needs given their resources. Regulations may also play a

role, e.g., it may be required that positive results are only declared based on an individual test. In this case, putative positives would require an individual test even if they were the only putative positive in a positive pool. In other settings, e.g., where the results will not be used diagnostically, the requirements may be more flexible, allowing labs to consider more sophisticated decoding methods that may save them more tests.

We have added some discussion of these aspects to the **Discussion (pg. 21)**.

This paper is serious mathematics, and appears to be accurate throughout. The paper is well written, with a thorough (perhaps even too thorough!) literature review. However, I have two main points of concern. First, I don't feel the results give me a good enough like-with-like comparison to other existing methods. Second, I'm not sure Nature Communications is the best venue for this paper to appear when (as it surely should be) it is published.

To give an example of my first point, take Figure 2(a) This compares an $n = 96$, $m = 16$, $q = 2$ HYPER design to an array design. I have a number of questions:

1) Why was an unbalanced 8×12 array chosen for comparison over a balanced 10×10 array with 4 "holes"?

We chose this specific array (and the corresponding 16×24 array for $n=384$) because it was an early, leading proposal in the literature for COVID-19 testing [9]. An important reason for choosing those sizes is that they are common plate sizes in laboratory settings, as are 8-channel and 12-channel pipettes. This real-life benefit is one of the key features of that proposal, since it simplifies the pipetting done during pooling.

That said, we agree that it is important to understand how a corresponding balanced array design would compare. We added a **new simulation (Supp Fig 5)** with remarks in the paper (**pg. 14**) that compares HYPER with a balanced 10×10 array having 4 holes for $n=96$. We also considered a 20×20 array with 16 holes for comparison with the $n=384$ designs. In both cases, the holes were placed along the diagonal to make the pools balanced. As with the unbalanced arrays, we found that HYPER was roughly 25% more efficient than the balanced arrays for much of the 50-day window. This is because the HYPER designs use fewer pools and hence have a lower number of tests that must always be run. HYPER also has correspondingly larger pool sizes, and is slightly less sensitive. Notably, HYPER designs that instead match the number of pools (**cf. Supp Fig 3**), appear to have very similar performance to these balanced arrays with holes.

2) Why was it not compared with a random $n = 96$, $m = 16$, $q = 2$ design?

Our focus in this paper was to develop a deterministic design. In the lab, one design must eventually be chosen to develop a fixed protocol that can be implemented. Designs that are (at least in theory) randomly drawn each time are more difficult to implement with robust quality control. Indeed, a major impetus for our present work came from the need to have a

deterministic design with analogous average performance to the random designs studied in our previous work (Cleary and Hay et al. 2021). Thus we focused on comparing with other deterministic proposals (array designs and P-BEST).

That said, we do note in the paper that the “average performance of the HYPER designs was generally consistent with those of the random assignment designs” (pg. 14). To help clarify this point, we have added a **new analysis (Supp Fig 5)** with remarks in the paper (pg. 14) that explicitly shows this comparison. Indeed, the average efficiency and the sensitivity are generally very similar.

However, the average behavior obscures some differences between this random design and HYPER. In particular, draws of the random design are often unbalanced with respect to both pools and pool combinations, whereas HYPER is always balanced in both ways. This is important in real life. For example, with imbalanced pools, the volumes to be drawn are different for each pool, making quality control more difficult in lab settings. Another important consequence is that the performance of random designs often depends on where positive individuals happen to fall in the design. We have added a **new analysis** investigating this (**Supp Fig 6**). It considers a single positive individual placed in each of the $n=96$ locations of the design. For several draws of the random design, we found that both the resulting efficiency and sensitivity depended on where the individual was. Some locations were more sensitive than others. HYPER on the other hand performed consistently regardless of where the individual was.

3) Why was it not compared with a random $n = 96$, $m = 16$, $q = 2$ design with the additional constraint that each pool contains 12 samples?

As before, we focused on comparing HYPER with other deterministic proposals because of the real-life importance of having a fixed design in the lab. That said, we agree that it would be illuminating to study how the average performance compares with a random design constrained to have balanced pools. The earlier random (assignment) design only guaranteed balance in the first sense (all individuals split into q pools). A random design with the additional constraint would add to this a guarantee of balance in the second sense (balanced pool sizes), allowing us to study the resulting impact of having balanced pools.

One random design satisfying the additional constraint is double-pooling [64]. This design randomly partitions all individuals into $m/2$ equally sized pools twice. Similar to the earlier random assignment design, we found that double-pooling had very similar average performance to HYPER (**Supp Fig 5**). Likewise, efficiency also depended on where the positive individual happened to fall (**Supp Fig 6**). However, since the pools are balanced in double-pooling, the sensitivity is uniform like HYPER.

4) Why was it not compared with a code-based design with the same parameters $n = 96$, $m = 16$, $q = 2$? (Perhaps no such code based design exists? Then what about a truncated code-based design?)

In general, there are many code-based designs one could consider. Since our primary focus is testing for COVID-19, we compared with P-BEST since it was one of the leading proposals in this context.

A P-BEST design with the same parameters could potentially be constructed, e.g., by some appropriate form of truncation, but this would be an extension of their proposal, which is beyond our scope to construct. Note that we would likely also have to retune their decoder for the new design, which might produce an unfair comparison if mistakenly done incorrectly. Rather than compare HYPER with our take on a modification of their proposal, we felt it was more fair to instead use their design as they provided. This is the comparison shown in the n=384 panel of **Fig 2**.

If a P-BEST design for n=96 becomes available, we would of course be more than happy to include it. We expect the comparison with HYPER will be similar to n=384.

Without such comparisons, how am I meant to know how good the HYPER design is?

We thank the Reviewer for raising this point, which we agree was important to consider. We feel that the comparisons with additional methods (**Supp Fig 5**) described above have improved and strengthened our manuscript. The broader question of how good HYPER is in general was also asked below. We give a more detailed response there.

Similarly, in the top-right (unlabelled) graph, we get a comparison with another un-balanced array and one particular code-based design (P-BEST) with completely different parameters. Unsurprisingly, the design with completely different parameters has completely different behaviour, but I don't see that the reader learns much from that.

As discussed above, the 16x24 array was chosen in the right panel of **Fig 2a** since it was one of the leading proposals for COVID-19 and is motivated by the real-life benefit of matching the standard plate sizes that are common in labs. The HYPER design shown was chosen to match the array design with respect to the largest pool size. P-BEST was included since it was also one of the leading proposals for COVID-19 testing. The three methods are unified in that they all screen n=384 individuals per batch.

That said, we agree that the comparison with P-BEST would be much more enlightening with HYPER designs that have more similar parameters. We have added a **new simulation (Supp Fig 4)** with remarks in the paper (**pgs. 13-14**) that compares P-BEST with two additional HYPER designs. One matches the number of pools (m) of P-BEST, and the other matches the pool sizes ($n \cdot q/m$) of P-BEST. For the HYPER design matching the number of pools of P-BEST, the efficiency at low prevalence is similar. However, its pools are one-third in size helping it achieve a higher sensitivity. The HYPER design matching the pool sizes of P-BEST has a comparable sensitivity to P-BEST for much of the 50-day window highlighted. However, it has one-third as many pools giving it an initial efficiency roughly three times higher. As before (**Fig**

2), the efficiency of HYPER declines for both designs as prevalence grows, eventually falling below the constant efficiency gain achieved by P-BEST around day 80. Likewise, as before, the sensitivity of HYPER grows around the same time, while P-BEST significantly loses sensitivity.

These results help us better understand which differences in behavior come from different design parameters (n,m,q) and which arise due to the one-stage approach and decoder of P-BEST in contrast to HYPER. For example, we see that even when the HYPER parameters are chosen to closely match P-BEST on either initial efficiency or initial sensitivity, the general impact of increasing prevalence is similar to before. HYPER loses efficiency but gains some sensitivity, because in its two-stage approach, more stage-one pools (correctly) test positive, requiring more tests in stage two. P-BEST (with the default parameters) loses sensitivity at higher prevalence but on the other hand maintains a constant efficiency gain. This has its own benefits, e.g., the workload is known beforehand. Labs must consider which form of degradation (in efficiency or sensitivity) is more tolerable based on their particular goals.

Subfigures 2b and 2c don't have any comparisons at all with other designs.

Indeed, since the flexibility of HYPER allows for many designs, our goal in these subfigures was to compare different HYPER designs and their various tradeoffs. We have clarified this in the paper (pg. 14). We found that the efficiency gain of HYPER was roughly n/m early in the simulated epidemic, during which pools frequently test negative due to low prevalence. Using fewer pools (m) can thus significantly increase efficiency but this generally came at the expense of a slight reduction in sensitivity. Moreover, the efficiency degraded as the prevalence grew, though we found that HYPER designs with more splits (q) were generally more robust to this loss in efficiency (while also being less sensitive).

I don't think the paper can be accepted without such comparisons to existing designs in the literature. How else is the reader meant to judge if the HYPER designs are good or not?

My suspicion is that the HYPER designs will have extremely good performance, which is why I'm positive on the paper as a whole. But I'm not sure whether they will be "slightly-but-undeniably better than random designs" or "loads better than random designs", and similarly versus code-based designs. I wish the paper had given me the information to know.

Since our primary focus is on testing for COVID-19, we felt it was most appropriate to focus our comparisons likewise on leading proposals in that context that (like HYPER) account for real-life considerations of implementation in the lab. Hence, the particular designs considered. Compared with those designs, we found that HYPER benefits greatly from its flexibility, which allows it to be more easily tailored and optimized for varying scenarios. Moreover, this flexibility did not appear to come at the expense of performance in the scenarios that were favorable for other methods (8x12 and 16x24 arrays and P-BEST).

However, we agree that additional comparisons (as proposed by the Reviewer) are important to better understand where the differences come from and how HYPER might compare with more

general group testing methods. As discussed above, we have added several new comparisons (**Supp Figs 3-5**). We thank the Reviewer for this suggestion, which we feel has made the comparisons more enlightening.

Compared to random designs, we found that HYPER (with matching parameters) generally had similar performance **on average (Supp Fig 5)**. However, **HYPER treated individuals in the design more uniformly (Supp Fig 6, new discussion on pgs. 14-15)**. The random designs are more variable and can be quite unbalanced for any given random draw. See our discussion on pitfalls of unbalanced designs in our response to point 4 of Reviewer 1's comments. Considering the case where one individual is positive, the average efficiency and sensitivity of the earlier random design depended on where the positive individual happened to fall in the design. Double-pooling likewise had uneven efficiency. HYPER, on the other hand, had uniform efficiency and sensitivity across all individuals. Hence, the stage-two workload is more consistent under HYPER, and a single positive individual has the same probability of being correctly identified regardless of where they happen to fall in the design. These are both very important for real-world testing.

Compared with the code-based one-stage design P-BEST, we found that similar behavior as observed before arose even for HYPER designs with more comparable parameters. For example, we considered matching the number of pools or matching the pool sizes.

To assess the performance of HYPER more generally, we have also added a new comparison (**Supp Fig 2**) with remarks in the paper (**pg. 11**) under the noiseless large n setting with fixed prevalence. In this setting, the recent work [A] has provided a bound on the best possible performance achievable by any conservative two-stage testing method. We found that HYPER appears to be fairly close to optimal across a wide range of prevalences. There remains a gap at small prevalence likely due to: a) looseness in our upper bound for HYPER, b) potential looseness in the lower bound [A], and c) our added constraint that $q \leq 3$ (to ease real-life implementation). Further sharpening our analysis of HYPER (especially for $q=3$) is an exciting direction for future work, and may show that HYPER is in fact even closer to optimal than our current results show. We thank the Reviewer for pointing us to the reference [A].

While we're here, the "exponentially increasing prevalence over time" model on the x-axis for the Figure was a bit too clever-clever for my taste – a simple range of plausible prevalences on the x-axis would be fine, without trying to shoehorn it into some purported pandemic model.

Supp Fig 1 considers a simpler model where performance is determined by prevalence alone (in addition to overall test sensitivity/specificity). In that setting, we can and in fact do show performance across a simple range of prevalences.

That said, the epidemic model is important for studying how the various methods would perform in real life. In particular, the effect of dilution is not easily captured by prevalence alone since it depends on not just how many individuals are positive but also on how much viral load each has. As a result, two days with the same prevalence can yield different behavior as a result of a

different viral load distribution at different points in epidemic time. This is perhaps most clearly seen in the new simulation of a two-wave epidemic (requested by Reviewer 3). There we find, e.g., that the sensitivity of the various methods is different on day 82 than day 104 as a result of differing viral load distributions (**Supp Fig 8**).

Second, we have the question of whether Nature Communications is the best venue for this paper. This is a lengthy paper (23 pages – or 49 pages with full supplementary information) that is quite mathematically technical. It presents evidence from computer simulations, rather than actual laboratory studies. I would normally recommend such a paper goes to a combinatorial journal – something like "Combinatorics, Probability and Computing" or "Discrete Applied Probability" or "Journal of Combinatorial Optimization". Understandably (and admirably!) the authors wish to advertise their web app to practitioners, but I'm not sure a long technical paper in Nature Communications is the best way to advertise this useful tool.

*If this paper had arrived in my inbox at one of those journals (and with appropriate comparisons with existing designs), I wouldn't hesitate to recommend acceptance. However, as we are, I narrowly come down on the side of rejection. If the editor feels - contrary to my thoughts above - that Nature Communications **is** in fact an appropriate venue for a paper such as this, I don't at all object, but would like some small revisions to see better comparisons with those existing designs before acceptance.*

Indeed, our goal with HYPER was to develop a method addressing some of the needs of practitioners (e.g., the need to have something simple, effective and flexible) in hopes that they may find it useful. As such, we made great efforts to present the method and key results in a reader-friendly form suitable for readers of Nature Communications, leaving many of the more technical details for the supplement.

In fact, some of the complexity and length comes from our attempt to account for major issues that arise in real life, such as sensitivity loss due to dilution, an evolving pandemic with shifting viral load distributions, and constraints on the sample collection and testing budget.

As such, we hope to reach a broad audience including practitioners with this work and feel Nature Communications would be an excellent venue for that. We are of course happy to make further modifications to help make the paper more accessible. Moreover, we also hope to follow up with more technical analyses in future works.

REFERENCES

[A] Aldridge, Conservative two-stage group testing, arXiv

[B] Zaman-Pippenger, Asymptotic analysis of optimal..., Prob Eng Inf Sci

[C] Kautz-Singleton, Nonrandom binary superimposed codes, IEEE Trans Inf Th

Response to Reviewer 3

Dear Editor Dr Righetto,

Thank you for asking me to review the research article "HYPER: Group testing via hypergraph factorization applied to COVID-19" by Hong D et al (NCOMMS-21-07610-T). As instructed, my comments are focused on the utility of the method and its potential of use among clinical practitioners, rather than the technical details.

Pooled testing of samples may help to alleviate the heightened demand for large-scale testing during pandemics, which is especially challenging for resource-limited facilities / areas. In this study, the authors proposed a new group testing method (HYPER) – a two-stage pooling strategy based on hypergraph factorizations. Using simulation studies, they predicted that HYPER may outperform other methods especially in resource-constrained environments, and required little expertise for implementation.

Strengths:

Important and timely question as pooled testing is a valuable method to help alleviate the burden of laboratories especially in resource-limited areas during pandemics; particularly as the COVID-19 pandemic may linger on for the foreseeable future until adequate herd immunity is developed.

We thank the Reviewer for their interest in our work, and for the thoughtful review and questions. We have addressed each of the questions and requests below, and are grateful to the Reviewer, as they have helped improve our manuscript.

Major comments:

1. Results (Performance under a COVID-19 model): the authors focused the simulation during which the prevalence increases from 0.03% to 2.46% (days 40-90 in the simulation); and applied individual testing sensitivity of ~85%. This reviewer considers that additional simulation with the following parameters to be especially relevant for clinical use:

i) Evaluating the post-exponential phase; ie: a later phase in the simulation. As evident by the COVID-19 pandemic, it is often difficult to have coordinated laboratory testing in the initial phase of pandemics. How would this additional simulation change the performance of HYPER and would this affect the perceived impact on the current COVID-19 pandemic which is now past the initial phase?

We thank the Reviewer for raising this important question, and agree that gaining an understanding of performance in a sustained epidemic, potentially with multiple waves, is critical for real-world applications. To address this point, in the revised manuscript we have included **new analysis** of a simulated population undergoing a sustained, two-wave epidemic.

In the **new analysis**, the simulated population (obtained from Cleary and Hay et al. 2021) was generated from an SEIR model with transmission rate modified at two time points. R_0 was initiated at 2.5 at day 0, decreased to 0.8 at day 80, and subsequently increased to 1.5 at day 150 (**new Supp Fig 8a**). The result is an epidemic with an initial wave, followed by a decline phase and subsequently another growth phase. We assessed changes in sensitivity, efficiency, and the relative performance of each method during each of these phases.

Sensitivity is generally lower for all methods (including individual testing) during the decline phase compared to the two growth phases. This is consistent with the findings of Cleary and Hay et al. (2021). Consider days 82, 104 and 173. These all have a prevalence of roughly 1.0% but the sensitivity on day 104 (which is during the decline phase) is generally lower than the other two days (which are during growth phases). This can be explained by observing that the distribution of nonzero viral loads (**new Supp Fig 8b**) on day 104 is shifted to the left relative to days 82 and 173, due to a shift away from recent infections in individuals sampled at random in the population. As a result, viral loads are more likely to fall below the limit of detection and get missed.

At the same time, the relative performance of the methods appears to be similar to our earlier simulation (**Fig 2**). The chosen HYPER designs are as sensitive as their corresponding array designs, while being roughly 20-25% more efficient for $n=96$, and roughly 10-25% more efficient for $n=384$. For $n=384$, the HYPER design is generally more efficient than P-BEST (up to 50% more) until day 180 while also being generally more sensitive. After day 180, P-BEST is more efficient but at an additional cost of sensitivity. As before, prevalence appears to be one of the main factors driving efficiency.

Overall, while the epidemic phases have different properties (e.g., due to differences in viral load distribution), the relative performance of HYPER designs appears to be fairly robust. This suggests similar overall conclusions for the current COVID-19 pandemic that is now past the initial phase. We have added a condensed version of this discussion to the paper (**pg. 15**).

That said, different scenarios will call for different choices of HYPER design. For example, the generally lower sensitivity during decline may be better handled by HYPER designs that use smaller pools and hence dilute samples less.

ii) Increasing the individual testing sensitivity to $\geq 90\%$. The gold standard of qRT-PCR used for diagnosing COVID-19 and other emerging viruses usually has sensitivity much higher than 85%. Therefore, the authors should show the simulation data using a higher individual testing sensitivity as well as determine the highest individual testing sensitivity which would make HYPER less desirable than individual testing.

We thank the Reviewer for raising this issue. We note that a sensitivity of around 85% appears to be roughly in line with recent studies for COVID-19 [75]. That said, we agree that it is important to understand how robust HYPER (and its benefits over individual testing) are to changes in test sensitivity.

As suggested, we have added a **new analysis (new Supp Fig 7)** using a higher individual testing sensitivity. In particular, the new simulation considers a lower limit of detection (LOD), reduced by a factor of 25 (from 100 to 4). A smaller LOD means that smaller viral loads can be caught, and as a result, the sensitivity of individual testing rises from roughly 85% to roughly 95%.

The sensitivity of HYPER (and the other group testing methods considered) is similarly increased in the new simulation. While sensitivity of all methods is increased, the difference in sensitivity between the pooling methods, and between pooling and individual testing, is somewhat unchanged. For example, averaged across days 40-90, the HYPER (n=96,m=16,q=2) design and the 8x12 array design have a 0.24 percentage point difference in sensitivity in the original simulation (**Fig 2**), and a 0.54 percentage point difference in the new simulation (**new Supp Fig 7**). In the new analysis, each of these designs continues to have lower sensitivity than individual testing due to the dilution of viral loads below the LOD during the pooled testing of stage one. However, the gap is now slightly smaller. For example, the HYPER (n=96,m=16,q=2) design had a sensitivity roughly 10 percentage points lower than individual testing for an LOD=100. For an LOD=4, this difference is roughly 9 percentage points. Note that in the extreme case of perfect individual testing sensitivity, HYPER will also have perfect sensitivity. All positives would make their pools positive, they would be declared putative positives and then identified in stage two.

Reducing the LOD appears to have had little effect on the efficiency (**new Supp Fig 7**). It is nearly the same as before (**Fig 2**), with HYPER enjoying essentially the same gains in efficiency over individual testing. In general an increase in sensitivity could lead to a slight potential decrease in efficiency, since more pools will (correctly!) test positive. However, this does not seem to have a big impact here. We have added a condensed version of this discussion to the paper (**pg. 15**).

Moreover, note that a higher sensitivity means that more aggressive pooling strategies (that sacrifice sensitivity to gain efficiency) become viable. For example, the HYPER (n=384,m=12,q=3) design has larger pools and is less sensitive than the HYPER (n=384,m=12,q=2) design but more efficient around day 60.

Based on these findings, we do not think there is a point at which increasing the sensitivity makes HYPER less effective than individual testing. In fact, under an assumption of noiseless testing (perfect sensitivity) our theoretical analysis shows that HYPER is at least 7 times more effective than individual testing up to a prevalence of 1% (**Supp Fig 1h, 2**).

2. Under the COVID-19 simulation model, the authors took into account a) SARS-CoV-2 viral load kinetics in infected patient, b) dilution of viral load during pooling, and c) evolution of infection prevalence in a large population over time. As HYPER may be a useful tool not only for COVID-19, but also future pandemics, can the authors provide simulation to predict how these parameters would affect the practicality of HYPER? For example:

We agree with the Reviewer's assessment of the importance of potential application in future pandemics, and thank them for highlighting this issue. As the Reviewer has laid out, we believe key considerations for prospectively evaluating pooling in these scenarios are changes in viral kinetics and epidemic dynamics, and the "interaction" of these to produce a distribution of viral loads at any given point in time. Although not modeled as an outbreak of any specific virus, we feel that the variety of analyses and modifications to our simulations that we have made allow us to address these concerns, as described in more detail below. Overall, the analysis indicates that viral load kinetics and dilution primarily impact sensitivity but do so similarly across all methods (including individual testing), such that HYPER remains among the most effective. Finally, we think a more slowly evolving prevalence would increase the practicality of HYPER since designs tailored for low prevalence can then be used for a longer time. We have added a discussion of these points to the **Results (pg. 15)** and the **Discussion (pg. 20)**.

a) Viral load kinetics: SARS-CoV-2 viral load tends to peak during the early phase of disease, while SARS-CoV-1 viral load peaks at around day 10. How would this different viral shedding pattern affect the performance of HYPER?

Viral load kinetics affect the efficiency and sensitivity of all the methods on any given day through the resulting day-specific distribution of viral loads in the population. Especially important is what fraction of positive viral loads are also above the LOD after dilution. A viral load distribution where this fraction is higher will have a correspondingly higher sensitivity since more positive cases will be caught in both stages. As a result, the efficiency is also slightly reduced since more positive cases get (correctly) caught in stage one, leading to potentially more tests in stage two. However, if prevalence is low, the group testing methods will still be significantly more efficient than individual testing since many negative cases will still be eliminated by the pooled tests.

In the specific case the Reviewer raises, with viral load peaking much later, the most important difference from SARS-CoV-2 might be that a larger proportion of false negatives will come from individuals who have been recently infected, but still have lower viral load. For infection control it is critical to identify and isolate positive individuals soon after infection to reduce the probability of transmission. Hence, the change in viral kinetics might result in more false negatives in these important cases. Moreover, the difference in sensitivity between growth and decline phases might be altered, since this difference arises from differing viral loads in the recently infected versus those late in infection.

However, the discussion above applies both for individual testing and pooled testing. As our **new analysis** (in response to the Reviewer's comment above) demonstrates, while a shifting distribution of viral loads does affect sensitivity and efficiency, it generally does so concordantly across methods, such that HYPER remains among the most effective in all scenarios. We have added a condensed discussion of this to the **Discussion (pg. 20)**.

b) Dilution of viral load during pooling: this would likely remain more or less constant.

The impact of dilution is driven primarily by: a) the amount of dilution (which is a property of the design), and b) the limit of detection (which determines the test sensitivity).

The impact of varying the amount of dilution was investigated in **Fig 2b**, which compared three HYPER designs with varying numbers of pools (and hence varying pool size and dilution). Increasing the pool size (by using fewer pools m for the same number of individuals n) generally yielded greater efficiency at low prevalence at the cost of a decrease in sensitivity. At high prevalence, the reverse happened, likely due to individuals with low viral load getting “rescued” by others in their pool. These general behaviors are likely to be the same for future pandemics.

To investigate the impact of varying the limit of detection, which could be different in future pandemics, we added **new analysis (new Supp Fig 7, discussed on pg. 15** and described in response to the Reviewer’s comment above regarding increased sensitivity). It repeats the earlier simulation (**Fig 2**) but with a 25-fold reduction in the limit of detection (LOD). A smaller LOD means that smaller viral loads can be caught. As a result, the sensitivity of individual testing rises to roughly 95%; the various group testing methods have a corresponding increase as well. The relative comparison of their sensitivities are similar to before. The efficiency for all the methods is also very similar to before. To summarize, the main impact was on sensitivity which improved across the board.

Put together, higher LODs generally lead to lower sensitivity (as viral loads more easily get diluted below the LOD). However, flexible methods like HYPER can adapt (in part) by tailoring the design to account for this, e.g., by using a design with smaller pools (diluting less). The flexibility of HYPER helps it remain practical. Note that smaller pools often come at the cost of efficiency, so in practice labs will need to consider how to properly weigh these tradeoffs according to their specific needs.

c) Evolution of infection prevalence in a large population over time: respiratory viruses such as coronaviruses tend to spread much faster than other emerging viruses transmitted by alternative routes (eg: fecal-oral or arthropod-borne). Would the longer time taken for the increase in prevalence of these alternative agents diminish the usefulness of HYPER?

We think the longer time taken for increase in prevalence would likely not diminish the usefulness of HYPER.

In fact, we think it would actually benefit HYPER. Essentially, if the prevalence remains low for a longer time, we can use HYPER designs that are very effective but only at low prevalence. These designs gain efficiency at low prevalence by sacrificing efficiency at high prevalence. For example, in our simulations (**Fig 2b**), designs that were the most efficient initially (i.e., at low prevalence) ended up being the least efficient later on (at high prevalence). An additional benefit of slowly changing prevalence is that the effectiveness of each design changes more slowly, making it easier to select a single design that remains effective over time.

The resulting impact on the overall effectiveness of HYPER can be seen from our simulations in **Fig 3** and **Supp Fig 10**. **Fig 3** investigates choosing optimally effective designs over a window of time during which the prevalence grows from 0.03% to 2.46%. **Supp Fig 10** considers a single day (day 53) during which the prevalence is 0.1%, and provides an idea of how things might look for a more slowly evolving infection prevalence.

Consider the testing-constrained setting shown in insert (a) of each figure. Under the growing prevalence (**Fig 3a**), HYPER achieved an effective screening capacity of 122.2 individuals with a design using batches of $n=192$ individuals, $m=6$ pools and $q=2$ splits. The chosen design must strike a balance between being effective early on (at low prevalence) and later on (at high prevalence). In contrast, for the fixed (or slowly evolving) prevalence of around 0.1% (**Supp Fig 10a**), HYPER reached an effective screening capacity of 231.6 individuals, a nearly two-fold improvement. It does so with a design ($n=384, m=8, q=2$) tailored for that prevalence, that can more fully exploit the fact that many individuals are negative. Indeed, HYPER (and group testing in general) typically benefits from low prevalence.

We have added a condensed discussion of this to the **Discussion (pg. 20)**.

3. Website (<http://hyper.covid19-analysis.org>): the number of subjects per batch (n) is limited between 6 and 300. Can the maximum n of 300 be increased further as many resource-limited areas have large populations which far exceeds 300 per batch (eg: single-source outbreak clusters of epidemiologically-linked COVID-19 cases of >300 patients have been well reported in different settings).

Absolutely, we have increased the maximum n to 6144.

We have also provided code to generate arbitrary HYPER designs here: <https://github.com/dahong67/hyper-group-testing/tree/main/hyper>

Note that large populations are sometimes more effectively screened under resource constraints by splitting them into multiple batches (b). In that case, the number of subjects per batch (n) will be a fraction of the total population being screened. As an example, the most effective HYPER design in Fig. 3f (among those considered) used $n=192$ subjects per batch with $b=1.9$ batches on average.

Reviewers' Comments:

Reviewer #1:

Remarks to the Author:

I thank the authors for incorporating most of my suggestions and providing the point-by-point list of responses to my questions. The paper is well structured and technically sound. Now it is well written and easy to read. I have checked their proofs and everything is correct for what I have been able to ascertain.

I have several small suggestions which might improve the presentation of the paper.

page 4. "For some scenarios,..." I believe that there is no need to write explicit real constants (like 3 and 14 you used) when comparing your design with other approaches. This statement has a vague meaning and doesn't properly reflect your contribution. I would avoid saying that.

page 8. Hypergraph factorization. It would be nice if you explicitly write that m is supposed to be divisible by q .

page 10. What does the overall average specificity/sensitivity mean? Before you recall the definition of the specificity of a test only.

page S2. Frankly speaking, I don't understand the purpose of discussing balanced array designs. In my opinion, this part can be safely removed without changing the value of the paper.

$z_{\{id\}}^{\{(pop)\}}$ or $z_{\{i,d\}}^{\{(pop)\}}$. Please check whether the notations are consistent.

page S12. Bottom. Please write $P(A_j \cap A_{\{j'\}})$ instead of $P(A_j \cup A_{\{j'\}})$. Also, three equality cases (A), (B), (C) are specified in the proof whereas the statement describes only two.

Reviewer #2:

Remarks to the Author:

I thank the authors for the revision to their paper and their detailed responses to reviewers, including myself. (I am "Reviewer 2" in their comments.)

SUMMARY: In my previous review I (narrowly) suggested rejection on two grounds: first, that there was insufficient evidence in the paper of the HYPER schemes outperforming existing schemes with the same parameters; and second, that this technical math paper without lab-work would be better suited to a combinatorics (or combinatorics-adjacent) journal. While this version contains improvements, my opinions are pretty much the same. On my first point, the new comparisons with existing schemes are welcome, and strengthen the paper – but I don't see clear and obvious benefits of HYPER over the established methods when the comparison is to schemes with the same parameters. (See point 5 below.) For the second point, I still think that the paper would be more welcome placed in front of readers who enjoy reading about "the projective line" and "orbits" and "primitive elements". The paper is mathematically solid – and now even more solid than before – so will be of interest to mathematicians for the style of its mathematical constructions, but the apparent lack of significant improvements over existing schemes with the same parameters may limit the interest of a wider non-mathematical audience.

Some more detailed comments, which I'll try to keep to just the new changes.

1) I'm pleased to see a clear formal definition of "maximally balanced", which is very welcome (although the mathematician in me was a bit sad that space couldn't be found for it in the non-supplementary parts of the paper).

2) I see from the authors' response that I confused matters in my earlier comments by using m in a different way to them. Apologies for my sloppiness. Let me try to do better this time: I'll use r instead, and keep m strictly to be the number of tests, as the authors do. I was using r to be the

size of the grid/array; that is an $r \times r$ grid tests r^2 items in $m = 2r$ tests. A single $r \times r$ grid is a maximally balanced design for $n = r^2$ items and $m = 2r$ tests. But also – and here is the point I failed to clearly make last time – a bunch of k disjoint $r \times r$ grids is also maximally balanced for $n = kr^2$ items and $m = 2kr$ tests. (I believe this may be – or perhaps is a special case of? – what the authors now later refer to as "double pooling"?)

Further, as I commented before and the authors agree in their response, "arrays with holes" are also maximally balanced under certain easy-to-sort-out circumstances (eg putting up to r holes in each array, for example down the diagonal).

All this is to say that the paper still does not seem, to me, to have a crystal clear answer to the basic question "When $q = 2$, what, if anything, is the benefit of HYPER over the simple and well-established array/double-pooling designs?" In their response, the authors try to flip the burden of proof on its head – "we do not see a performance benefit to using array designs [over HYPER]" [response, p18] – but that doesn't really wash with me.

3) I'm a bit confused by the authors' justification, in a few places, of using non-square arrays, which thus suboptimally provide tests of two different sizes (eg the 12×8 array has 12 tests of size 8 and 8 tests of size 12) over sticking to square arrays (eg the 10×10 array has all 20 tests of size 10; or 12 of size 10 and 8 of size 9, if you punch four holes in it). The authors make comments about "plate sizes" and pipettes. But the "array" is, of course, merely a conceptual crutch for explaining mathematically the design of the pools, and need not correspond to an actual physical "array" in a laboratory somewhere. Unless my ignorance of lab-work is leading me to misunderstand how this works in practice?

4) I'm pleased the authors have clarified the conservative/nonconservative (aka trivial/nontrivial) issue. I agree totally with the authors' defense of the conservative framework.

5) In my original review, I posed a number of questions about comparisons of HYPER with existing schemes *that have the same parameters as the HYPER variant under consideration*. (By "parameters" I mean number of items, number of tests, number of tests for each item, and number of items in each test (if constant).) This last point is absolutely crucial: comparing schemes with different parameters does not tell us about the quality of the the schemes themselves, merely about the appropriateness of the chosen parameters. Thus I am very pleased to see the comparisons in Supplementary Figures 3 and 5 which are entirely welcome. Indeed, for me, these Figures are the most important part of the paper beyond the central definitions, so it seems a shame to have them shunted off into the supplementary material. Unfortunately, the Figures appear to show that their new HYPER scheme performs about the same as the simple random, array, or double pooling schemes, which is a shame. Thus, while these simulations have undoubtedly strengthened the scientific rigour of the paper, they don't seem to have improved the impressiveness of HYPER over existing schemes. (Except, of course, to mathematicians who would be impressed by the details of the HYPER construction quite apart from its performance in simulations of Covid testing.)

6) Following from my point 5), I'm disappointed the authors did not add a comparison of HYPER with a code-based scheme (following the Kautz-Singleton construction) with the same parameters. Their excuse for not doing so appears to be that a previous paper that described one particular code-based scheme, called P-BEST, didn't use the authors' parameters – see the authors' comment claiming it would not be in appropriate to use "our take on a modification of their [P-BEST's] proposal" [response, p19]. This seems to have the horse and cart the wrong way around: the Kautz-Singleton framework has been around for 57 years, and the recent P-BEST paper discusses one microscopic special case of it. Refusing to take a more appropriate case of the Kautz-Singleton design because it would be "modifying P-BEST" seems unconvincing to me.

7) I found the new Supplementary Figure 6 interesting. This seems to suggest a "fairness" argument for preferring HYPER (or, in the second case, double pooling) over purely random designs, even at similar mean performance. It would be interesting to see how code-based designs fare on this measurement.

Reviewer #3:

Remarks to the Author:

The authors have performed new analyses as suggested. Their responses to the points raised in the previous round of review are considered well addressed by this Reviewer.

Overview

We thank the editor and reviewers for their thoughtful and constructive suggestions, which have helped us significantly strengthen the paper. Notably, we expanded our comparison of HYPER with existing methods and found that **HYPER does indeed have clear advantages over broad classes of existing gold standard methods using the same set of design parameters as HYPER.**

For your convenience, we have copied the reviews below (in *black italics*) with our responses provided point-by-point (in blue). References made (e.g., figure numbers) in the reviews were left as is; references in our responses correspond to the revised version.

We also indicate changes in the manuscript in blue.

To briefly highlight some key points of our revisions:

- We added **new and expanded comparisons of HYPER with the broad class of balanced arrays (Supp Figs 17-18)**, where both were optimized over the same set of design parameters (**Supp Table 1**). HYPER is about as effective as or substantially more effective than balanced arrays across the entire grid of resource constraints. Moreover, in important testing-constrained scenarios (with sample collection significantly outstripping testing capacity), **HYPER is up to 38% more effective than balanced arrays.**

We have added new discussion of these results on **pgs. 19-20** and accordingly updated the discussion on **pgs. 22-23**.

- We added **new comparisons of HYPER with the broad class of Reed-Solomon Kautz-Singleton (RS-KS) code-based designs (Supp Figs 19-20)**, where both were optimized over the same restricted subset of design parameters allowed by RS-KS (indicated in **Supp Table 1** by asterisks). **HYPER is up to 34% more effective than RS-KS** in important testing-constrained settings and is about as effective as RS-KS otherwise.

We have added a new discussion of these results on **pgs. 19-20** and accordingly updated the discussion on **pgs. 22-23**.

- We clarified **that random designs (random assignment and double-pooling) often have unbalanced pools or pool combinations, resulting in variable performance (Supp Figs. 5 and 7; discussion on pgs. 14-15)**. In contrast, HYPER guarantees that the pools and pool combinations are always maximally balanced. It had consistent sensitivity and efficiency regardless of which individual was positive. Variable sensitivity and workload are undesirable from a laboratory standpoint, making HYPER superior to both these random designs for any parameters.

- We added **balanced array designs and RS-KS code-based designs to the analysis of balance and performance consistency (Supp Fig 7)**. We found that the balanced array design and the RS-KS design both had uniform sensitivity and slightly uneven efficiency.

We have updated the discussion on **pg. 15**.

- We provided a couple additional **new comparisons of efficiency/sensitivity over time with illustrative balanced array and RS-KS designs (new Supp Fig 6** discussed on **pg. 14**). We found that HYPER designs (with matching parameters) are more efficient in some cases.
- We added **new revisions** that: clarify overall average sensitivity/specificity (**pg. 10**), describe the construction of general balanced array designs (**Supp Methods, pgs. S2-S3**), and describe the construction of RS-KS designs (**Supp Methods, pgs. S3-S6**). We also addressed all other suggested edits from Reviewer 1.

We hope we have adequately addressed all comments and look forward to hearing from you.

Response to Reviewer 1

I thank the authors for incorporating most of my suggestions and providing the point-by-point list of responses to my questions. The paper is well structured and technically sound. Now it is well written and easy to read. I have checked their proofs and everything is correct for what I have been able to ascertain.

Thank you again for your questions, comments and suggestions, which helped improve and strengthen the paper.

I have several small suggestions which might improve the presentation of the paper.

page 4. "For some scenarios,..." I believe that there is no need to write explicit real constants (like 3 and 14 you used) when comparing your design with other approaches. This statement has a vague meaning and doesn't properly reflect your contribution. I would avoid saying that.

Agreed, done.

page 8. Hypergraph factorization. It would be nice if you explicitly write that m is supposed to be divisible by q .

Agreed, done.

page 10. What does the overall average specificity/sensitivity mean? Before you recall the definition of the specificity of a test only.

Thanks for raising this question. We use "overall average specificity/sensitivity" to refer to the specificity/sensitivity of the overall procedure.

Namely,

- the overall average specificity is the probability a sample is declared negative by HYPER given that it was negative, and
- the overall average sensitivity is the probability a sample is declared positive by HYPER given that it was positive,

where the probabilities are with respect to the random test errors and the random positivity of the other individuals.

We have clarified this in the paper (**pg. 10**).

page S2. Frankly speaking, I don't understand the purpose of discussing balanced array designs. In my opinion, this part can be safely removed without changing the value of the paper.

These descriptions are needed to help clarify our comparisons with these balanced array designs. In particular, Reviewer 2 asked for a detailed comparison with these designs.

$z_{\{i,d\}^{\{pop\}}}$ or $z_{\{i,d\}^{\{pop\}}}$. Please check whether the notations are consistent.

Thanks, done.

page S12. Bottom. Please write $P(A_j \cap A_{j'})$ instead of $P(A_j \cup A_{j'})$. Also, three equality cases (A), (B), (C) are specified in the proof whereas the statement describes only two.

Thanks, done: changed to cap and added equality case (C) to statement.

Response to Reviewer 2

I thank the authors for the revision to their paper and their detailed responses to reviewers, including myself. (I am "Reviewer 2" in their comments.)

SUMMARY: In my previous review I (narrowly) suggested rejection on two grounds: first, that there was insufficient evidence in the paper of the HYPER schemes outperforming existing schemes with the same parameters; and second, that this technical math paper without lab-work would be better suited to a combinatorics (or combinatorics-adjacent) journal. While this version contains improvements, my opinions are pretty much the same. On my first point, the new comparisons with existing schemes are welcome, and strengthen the paper – but I don't see clear and obvious benefits of HYPER over the established methods when the comparison is to schemes with the same parameters. (See point 5 below.) For the second point, I still think that the paper would be more welcome placed in front of readers who enjoy reading about "the projective line" and "orbits" and "primitive elements". The paper is mathematically solid – and now even more solid than before – so will be of interest to mathematicians for the style of its mathematical constructions, but the apparent lack of significant improvements over existing schemes with the same parameters may limit the interest of a wider non-mathematical audience.

Thank you very much for your thoughtful comments and constructive suggestions, especially regarding how to more convincingly demonstrate benefits of HYPER over existing methods.

Based on your comments, we have expanded our earlier comparisons and identified important scenarios where HYPER does in fact outperform existing methods with the same parameters. Taken with our earlier comparisons, we have thus identified the following high-level advantages of HYPER over the following established methods:

- **Random designs (random assignment & double-pooling):** HYPER had substantially more consistent performance than random assignment and double-pooling, which is beneficial for laboratory implementation (**Supp Fig 7** discussed on **pgs. 14-15**).

We discuss this briefly in our response to point 2 and elaborate on it in our response to point 5.

- **Balanced array designs:** HYPER is up to 38% more effective than balanced arrays with the same design parameters in important testing-constrained scenarios (**Supp Figs 17-18** discussed on **pgs. 19-20**).

We discuss this briefly in our response to points 2-3 and elaborate on it in response to point 5.

- **Reed-Solomon Kautz-Singleton (RS-KS) code-based designs:** HYPER is up to 34% more effective than RS-KS in important testing-constrained scenarios, even when

HYPER is restricted to use the same design parameters as those allowed by RS-KS designs (**Supp Figs 19-20** discussed on **pgs. 19-20**).

We elaborate on this in our response to point 6 below.

We believe the new revision: **1) demonstrates clear benefits of HYPER over established methods with the same parameters, and 2) is consequently of interest to the wider non-mathematical audience.** We thank you for your detailed and thoughtful comments that encouraged us to expand our comparisons and clearly identify/demonstrate these benefits. We feel the paper is now much stronger as a result.

In addition to elaborating on the above comparisons, our responses below provide clarifications regarding designs formed by disjoint copies (response to point 2) and lab implementations of array designs (response to point 3). We also expanded our analysis of the impact of imbalance as suggested to include balanced array designs and RS-KS designs (response to point 7).

Some more detailed comments, which I'll try to keep to just the new changes.

1) I'm pleased to see a clear formal definition of "maximally balanced", which is very welcome (although the mathematician in me was a bit sad that space couldn't be found for it in the non-supplementary parts of the paper).

Thank you for your earlier suggestion to add it. If the editor feels it would be more appropriate in the main body, we will happily move it up.

2) I see from the authors' response that I confused matters in my earlier comments by using m in a different way to them. Apologies for my sloppiness. Let me try to do better this time: I'll use r instead, and keep m strictly to be the number of tests, as the authors do. I was using r to be the size of the grid/array; that is an $r \times r$ grid tests r^2 items in $m = 2r$ tests. A single $r \times r$ grid is a maximally balanced design for $n = r^2$ items and $m = 2r$ tests. But also – and here is the point I failed to clearly make last time – a bunch of k disjoint $r \times r$ grids is also maximally balanced for $n = kr^2$ items and $m = 2kr$ tests. (I believe this may be – or perhaps is a special case of? – what the authors now later refer to as "double pooling"?)

Thanks for the clarification. Indeed, k disjoint $r \times r$ grids taken together produce a maximally balanced design for $n = kr^2$ items and $m = 2kr$ tests.

In fact, this corresponds to what we called "batches" in the section "Choosing a pooling method given resource constraints" (**pg. 16**). There, each design was run for b batches, which is equivalent to forming a new design by combining b disjoint copies of the "base" design.

More precisely, for $r \times r$ grids, the following are equivalent:

- A. running b batches of an $r \times r$ grid that covers $n = r^2$ items / batch with $m = 2r$ tests / batch, and

B. doing a single run of a design formed by combining b disjoint $r \times r$ grids that covers $n = br^2$ items with $m = 2br$ tests.

(Note that “double pooling” is something different - it is a random design formed by randomly partitioning all n individuals into $m/2$ pools twice.)

As a result, **our comparisons under resource constraints (including the new comparisons discussed below) in fact already included this type of design.**

We have revised the paper to clarify this point (pg. 20).

Note also that the efficiency gain (# individuals screened / # average number of tests used, including stage-two tests) and average sensitivity (fraction of positive individuals identified after completion of both stages) are essentially the same for k disjoint $r \times r$ grids as for a single $r \times r$ grid. Thus, the comparisons done in “Performance under a COVID-19 model” for $k=1$ also apply to $k > 1$.

Further, as I commented before and the authors agree in their response, “arrays with holes” are also maximally balanced under certain easy-to-sort-out circumstances (eg putting up to r holes in each array, for example down the diagonal).

All this is to say that the paper still does not seem, to me, to have a crystal clear answer to the basic question “When $q = 2$, what, if anything, is the benefit of HYPER over the simple and well-established array/double-pooling designs?” In their response, the authors try to flip the burden of proof on its head – “we do not see a performance benefit to using array designs [over HYPER]” [response, p18] – but that doesn’t really wash with me.

Thank you for raising this question.

As hinted above, our new analysis demonstrates that **HYPER does in fact have important benefits over both balanced array designs (i.e., “arrays with holes”) and double-pooling designs.**

Namely, we found that:

- In important low-prevalence testing-constrained scenarios, **HYPER was significantly more effective than the array designs** with the same design parameters (**Supp Figs 17-18** discussed on **pgs. 19-20**).
- **HYPER had substantially more consistent efficiency than double-pooling**, which is beneficial from a laboratory standpoint (**Supp Fig 7** discussed on **pgs. 14-15**).

We elaborate on both these points in our response to point 5 below.

3) I'm a bit confused by the authors' justification, in a few places, of using non-square arrays, which thus suboptimally provide tests of two different sizes (eg the 12×8 array has 12 tests of size 8 and 8 tests of size 12) over sticking to square arrays (eg the 10×10 array has all 20 tests of size 10; or 12 of size 10 and 8 of size 9, if you punch four holes in it). The authors make

comments about "plate sizes" and pipettes. But the "array" is, of course, merely a conceptual crutch for explaining mathematically the design of the pools, and need not correspond to an actual physical "array" in a laboratory somewhere. Unless my ignorance of lab-work is leading me to misunderstand how this works in practice?

Lab implementations of array designs **do** in fact use actual physical arrays/plates to form the pools. This makes it possible to save pipetting steps by using multi-channel pipettes, which simultaneously pipette multiple wells (along a line) in a single step. In the lab, these savings can significantly speed up the overall procedure (forming pools accounts for a significant portion of the overall time). In these implementations, **the array is not merely a conceptual tool** for describing the mathematical design of the pools.

That said, HYPER lacks this feature since its pools do not neatly fall along lines of a grid. Thus, we agree that it makes sense to also compare with balanced square arrays with holes, where the array is now treated as merely a conceptual tool for constructing the design. Our new analysis (see response to point 5 below) does a thorough comparison with these designs.

4) I'm pleased the authors have clarified the conservative/nonconservative (aka trivial/nontrivial) issue. I agree totally with the authors' defense of the conservative framework.

Thank you for your earlier questions, which helped us clarify these points (for ourselves too!).

*5) In my original review, I posed a number of questions about comparisons of HYPER with existing schemes *that have the same parameters as the HYPER variant under consideration*. (By "parameters" I mean number of items, number of tests, number of tests for each item, and number of items in each test (if constant).) This last point is absolutely crucial: comparing schemes with different parameters does not tell us about the quality of the the schemes themselves, merely about the appropriateness of the chosen parameters. Thus I am very pleased to see the comparisons in Supplementary Figures 3 and 5 which are entirely welcome. Indeed, for me, these Figures are the most important part of the paper beyond the central definitions, so it seems a shame to have them shunted off into the supplementary material. Unfortunately, the Figures appear to show that their new HYPER scheme performs about the same as the simple random, array, or double pooling schemes, which is a shame. Thus, while these simulations have undoubtedly strengthened the scientific rigour of the paper, they don't seem to have improved the impressiveness of HYPER over existing schemes. (Except, of course, to mathematicians who would be impressed by the details of the HYPER construction quite apart from its performance in simulations of Covid testing.)*

Thank you for raising this point. As mentioned above, we have revised the paper and expanded the comparisons to more clearly demonstrate that HYPER does in fact outperform the random designs (random assignment / simple random and double-pooling) and balanced array designs, where the comparisons are done using the same design parameters.

Advantages of HYPER over random assignment (simple random) and double-pooling

In our previous analysis, the performance of random assignment and double-pooling designs were indeed similar *on average* to that of HYPER when they had the same design parameters (as was previously noted on **pg. 14** and shown in **Supp Fig. 5**).

However, **HYPER performed more consistently than both the random assignment and double-pooling designs using the same parameters (Supp Fig 7)**. Random assignment designs often have imbalanced pools and pool combinations, resulting in uneven efficiency and sensitivity. Likewise, double-pooling designs often have imbalanced pool combinations and uneven efficiency. HYPER guarantees that pools and pool combinations are both maximally balanced; it had both uniform sensitivity and efficiency.

Variable sensitivity and efficiency are undesirable from a laboratory standpoint, making HYPER superior to these designs even when they have the same average performance. Consistent sensitivity is important for real-world testing since all individuals should receive the same quality of treatment. Likewise, consistent efficiency produces more consistent workload, helping labs to better plan and efficiently allocate tests. In our experience talking with scientists who run testing labs, we have found that these aspects matter for real-world testing.

One could potentially attempt to achieve balance for random assignment or double-pooling by randomly generating a few designs, then trying to modify the “most balanced” one by hand to achieve balance. However, such an approach is ad-hoc and may still not result in a maximally balanced design. In contrast, HYPER produces a maximally balanced design directly. Note that this applies for any choice of the design parameters.

Advantages of HYPER over balanced array designs

In our previous analysis, we found that the average performance of a balanced array (i.e., a “square array with holes”) was indeed similar to HYPER *for some choices of design parameters* (cf. the “10x10 array with 4 holes” in **Supp Fig. 5** and the “HYPER($n=96, m=20, q=2$)” design in **Supp Fig. 3**).

Our new analysis broadens our previous analysis to now consider design parameters from the entire sweep we considered for HYPER. Namely, **we compared the performance of HYPER and balanced array designs under resource constraints, where both were optimized over the set of parameters from Supp Table 1.**

For prevalences of 0.1% and 1.06% (**Supp Figs 17-18**), we found that:

- **HYPER was about as effective as or substantially more effective than balanced arrays** under all sets of resource constraints.
- **HYPER was up to 38% more effective than the balanced arrays** in some of the most important scenarios (i.e., testing-constrained settings with sample collection capacity in significant excess to testing capacity).

We have added a discussion of these new findings to the paper (pgs. 19-20).

To elaborate on the second point here, for a prevalence of 0.1% and a budget of 1536 samples and 24 tests per day, HYPHER had an effective screening capacity of 624.1 individuals per day (Supp Fig 17b, copied below for your convenience). This is a roughly **25% improvement over the balanced array design with the same parameters**, which achieved 496.9 effective individuals screened. The best design parameters for both methods was the same ($n=768, m=12, q=3$), but HYPHER was more efficient (recall that for this resource analysis, we scan all designs that can be accommodated by a given set of resource constraints/budget, and identify the design resulting in the greatest number of effective individuals screened).

For a budget of 384 samples and 12 tests per day, HYPHER had a roughly **38% larger effective screening capacity** than the balanced arrays (231.6 vs. 167.7 effective individuals screened, Supp Fig 17a, copied below for your convenience). In this case, the greater efficiency of HYPHER made it possible to use more aggressive design parameters ($n=384, m=8, q=2$) within the resource constraints. The balanced array design with the same parameters was less efficient (see Supp Fig 6a and discussion on pg. 14), needed more stage-two tests, and failed to satisfy the testing constraint. A similar comparison occurred for a prevalence of 1.06% and a budget of 96 samples and 12 tests (Supp Fig 18a, copied below for your convenience).

Note that the balanced array designs used in the new analysis go beyond those we previously considered in that the new analysis includes designs with $n > (m/2)^2$ as well as $q=3$ “cube arrays” (for detailed descriptions of the constructions see Supp Methods, pgs. S2-S3). By construction, they have the same number of items (n), the same number of tests (m), and the same number of tests for each item (q) as their corresponding HYPHER designs. Furthermore, they have maximally balanced pools so have the same number of items in each test (nq/m) as their corresponding HYPHER designs in the cases where that number is constant across pools.

Recall also that, as discussed in our response to point 2 above, this resource analysis includes designs formed by combining multiple disjoint copies of a balanced $r \times r$ array design. Moreover, note that the most effective array designs in the analysis often had $n > (m/q)^q$. A larger number

of individuals per batch helps these designs exploit low prevalence. HYPER outperforms the usual array designs that are limited to $q=2$ and $n \leq (m/2)^2$ even more significantly.

6) Following from my point 5), I'm disappointed the authors did not add a comparison of HYPER with a code-based scheme (following the Kautz-Singleton construction) with the same parameters. Their excuse for not doing so appears to be that a previous paper that described one particular code-based scheme, called P-BEST, didn't use the authors' parameters – see the authors' comment claiming it would not be appropriate to use "our take on a modification of their [P-BEST's] proposal" [response, p19]. This seems to have the horse and cart the wrong way around: the Kautz-Singleton framework has been around for 57 years, and the recent P-BEST paper discusses one microscopic special case of it. Refusing to take a more appropriate case of the Kautz-Singleton design because it would be "modifying P-BEST" seems unconvincing to me.

While P-BEST was one of the leading code-based proposals for COVID-19 testing, it is indeed a single instance of the more general class of code-based schemes. As suggested, we have added a comprehensive comparison with code-based schemes following the Kautz-Singleton construction. Thank you for this valuable suggestion.

In particular, we compared the performance of HYPER and Kautz-Singleton designs based on Reed-Solomon codes (RS-KS, described in **Supp Methods, pgs. S3-S6**) using the same set of design parameters we previously considered for HYPER (**Supp Table 1**).

We found that:

- RS-KS was limited to a restricted subset of the design parameters.
- HYPER was more effective than RS-KS, even when restricted to only use the same subset of parameters as allowed by RS-KS.

Namely, the results demonstrate that **HYPER outperforms RS-KS with the same parameters**.

Restricted subset of parameters allowed by RS-KS

The standard presentation of RS-KS designs limits the number of individuals to $n \leq (m/q)^f$ (f is an additional design parameter, namely when considering $RS(q,f)$ codes over an alphabet of size b , $f-1$ is the degree of the generating polynomial; and moreover $m/q=b$ comes from moving to binary codes in the KS construction). However, this limitation is straightforwardly addressed by recycling pool assignments. Namely, one assigns individual $(m/q)^f+1$ to the same pools as individual 1, individual $(m/q)^f+2$ to the same pools as individual 2, and so on (see discussion on **pg. S5**).

Nevertheless, **the RS-KS construction remains limited to a restricted subset of the design parameters previously considered for HYPER** (denoted by asterisks in **Supp Table 1**). This is because RS-KS requires that

- **m/q is a prime power**

(m/q=b is the size of the alphabet in the RS code, which needs to be the size of a finite field in the default construction)

- **m/q >= q, i.e., m >= q^2**

(m/q=b, and q<=b is needed so that the evaluation map in the polynomial construction of the RS code will be at distinct elements of the field)

These constraints essentially arise in the construction of the (m/q)-nary Reed-Solomon code and appear to be difficult to remove without losing balance in the design (**pgs. S4-S5**). In particular, one could try to “puncture” the code in various ways or consider various sub-matrices of the RS-KS design, but it is unclear how to do so while preserving maximal balance. Going beyond the natural approaches we considered may be an interesting direction for future research but is beyond our present scope.

Advantages of HYPER over RS-KS even when restricted to parameters allowed by RS-KS

To compare HYPER and RS-KS, we investigated their performance under resource constraints.

Since RS-KS is limited to a restricted subset of design parameters (indicated by asterisks in **Supp Table 1**), we compared RS-KS (optimized over the restricted subset of parameters) with:

- A. HYPER optimized over the full set of parameters (i.e., all parameters in **Supp Table 1**)
- B. “Restricted HYPER” which is optimized over only the restricted subset of parameters

Namely, **Restricted HYPER is optimized over the same set of parameters as RS-KS.**

We found that HYPER has similar advantages over RS-KS as it had over balanced arrays (see response to point 5 above). Namely, for prevalences of 0.1% and 1.06% (**Supp Figs 19-20**):

- **HYPER and Restricted HYPER were both about as effective as or substantially more effective than RS-KS** under all sets of resource constraints.
- **HYPER and Restricted HYPER were both up to 34% more effective than RS-KS** in some of the most important scenarios (i.e., testing-constrained settings with sample collection capacity in significant excess to testing capacity).

We have added a discussion of these new findings to the paper (**pgs. 19-20**).

To elaborate here on the second point (as we did in our response to point 5), consider the important low-prevalence testing-constrained scenario of a prevalence of 0.1% and a budget of 1536 samples and 24 tests per day. In this case, HYPER and Restricted HYPER had an effective screening capacity of 624.1 individuals per day (**Supp Fig 19b**, copied below for your convenience). This is a roughly **29% improvement over the RS-KS design having the same parameters**, which achieved 483.8 effective individuals screened. Similar to our comparisons with balanced arrays, the best design parameters for all three methods were the same here (n=768,m=12,q=3) but HYPER was more efficient.

For a budget of 384 samples and 12 tests per day, HYPHER and Restricted HYPHER had a roughly **34% larger effective screening capacity** than RS-KS (231.6 vs. 172.4 effective individuals screened, **Supp Fig 19a**, copied below for your convenience). Similar to our comparison with balanced arrays, the greater efficiency of HYPHER made it possible to use more aggressive design parameters ($n=384, m=8, q=2$) without violating the resource constraints. The RS-KS design with the same parameters was less efficient (see **Supp Fig 6b** and discussion on **pg. 14**), needed more stage-two tests, and failed to satisfy the testing constraint. A similar comparison occurred for a prevalence of 1.06% and a budget of 96 samples and 12 tests (**Supp Fig 20a**, copied below for your convenience).

Recall that, as discussed in our response to point 2 above, this resource analysis also includes designs formed by combining multiple disjoint copies of an RS-KS design. Moreover, note that the most effective RS-KS designs in the analysis had $n > (m/q)^f$. A larger number of individuals per batch helps these designs exploit low prevalence. HYPHER outperforms the standard variant of RS-KS (that is limited to designs with $n \leq (m/q)^f$) even more significantly.

7) I found the new Supplementary Figure 6 interesting. This seems to suggest a "fairness" argument for preferring HYPHER (or, in the second case, double pooling) over purely random designs, even at similar mean performance. It would be interesting to see how code-based designs fare on this measurement.

Indeed, this is an important point (as discussed in our response to point 5 above). Consistent efficiency and sensitivity are desirable from a laboratory standpoint and were an important motivation for constructing maximally balanced designs (see **pgs. 4-5**). **Between two designs having similar average performance, the design with the more consistent performance is preferable.** Consistent efficiency, e.g., is an important benefit of HYPHER over double-pooling.

As suggested, we have added a code-based design to the figure (now **Supp Fig. 7**). For the parameters considered there ($n=96$ individuals, $m=16$ pools, $q=2$ splits), the Reed-Solomon Kautz-Singleton (RS-KS) code-based design had balanced pools and was maximally balanced

on the $(m/2)^2$ pool combinations it used. However, it was not perfectly balanced on the pool combinations it used; some were used twice while others were used once.

We also added the corresponding balanced array design. Like the RS-KS design, it had balanced pools and was maximally but not perfectly balanced on the $(m/2)^2$ pool combinations it used.

As a result, **both the RS-KS design and the balanced array design had uniform sensitivity and slightly uneven efficiency.** Note that their efficiencies were more uniform than random assignment and double-pooling, which do not even guarantee maximal balance on the pool combinations used. Note also that restricting them by requiring $n \leq (m/2)^2 = 64$ would make them balanced on the pool combinations they use. In contrast, HYPER had uniform efficiency here; it remains perfectly balanced on the pool combinations used for n up to $\binom{m}{q} = 120$.

The overall efficiencies of the RS-KS design and the balanced array were also lower than HYPER because they both assigned some pool combinations more than once. HYPER uses all $\binom{m}{q} = 120$ available pool combinations so each individual was assigned a unique pool combination.

Thank you for the suggestion.

Response to Reviewer 3

The authors have performed new analyses as suggested. Their responses to the points raised in the previous round of review are considered well addressed by this Reviewer.

Thank you again for suggesting the new analyses, which helped improve and strengthen the paper.

Reviewers' Comments:

Reviewer #4:

Remarks to the Author:

This paper considers the problem of group testing (a.k.a. pooled testing). This is a method of making medical testing more efficient at a time of constrained test availability, by testing samples together in pools and (for example) ruling out all the samples in a pool which together tests negative. Due to the COVID pandemic, this is a particularly topical problem, and has been applied in a variety of circumstances. However, this has generally involved Dorfman-style pooling ('partition the population into distinct pools, if a pool is positive retest each member individually') and because of the parallel nature of PCR testing for example there is interest in designing test strategies which can proceed non-adaptively, or at least in a small number of stages, particularly in the presence of test error.

This paper achieves this by a test design based on so-called hypergraph factorization. To the best of my knowledge this is a novel strategy, and the simulations presented show that it performs well against a variety of standard test strategies. In that sense the work seems interesting and valuable and from looking at the past referees' reports I am inclined to say that the issues raised there have been properly dealt with. (For clarity, I should say that I have not reviewed previous rounds of submission of this paper. I was asked to take over from the previous Reviewer 2 and to provide a mathematical perspective on this work, but if my comments appear inconsistent with previous reviews or suddenly ask for things at a late stage then this is the reason).

However, I have the following additional comments:

1. Although I am happy with the level of mathematical detail, I am not sure that it is explained in the right order. Specifically, on P6, the term 'hypergraph factorizations' is used, but this isn't defined until P9. I think it would make sense to restructure the material of P6-9, to first define what a hypergraph is (this may not be familiar to the general readership of this journal, particularly those from a medical background interested in group testing), then explain what is meant by a hypergraph factorization (ideally linked to an example), then explain how it is possible to use this to create a pooling strategy, and then to explain how the infection status of individuals can be deduced from this.
2. Related, I'm not sure that it's worth saying (as on P5) that here's a strategy that we {\em don't} employ -- i.e. cycling through pairs. Personally I'd prefer to explain the strategy (even in high level terms) that is used, as soon as possible in the paper.
3. While a comparison with alternative methods is made in Figure 2, and individual testing is a reasonable benchmark for this, I wonder whether it would be worth plotting the performance of standard Dorfman testing (with optimized pool size), at least in the noiseless case. That is, since the authors essentially consider a two-stage strategy (with conservative strategy of individual tests of those whose status is not clear), then it feels to me that Dorfman is a fair additional comparator. (Incidentally, it seems somewhat overcomplicated to plot the x axis of this as daily prevalence generated under an epidemic model -- it seems fine to me to plot simply prevalence directly, perhaps on a logarithmic scale).
4. In terms of decoding, even if it's not possible to prove anything formally, is there anything to be said by simulation about the performance of more intricate algorithms such as DD (in the language of [34]). It seems to me that this strategy is essentially COMP (rule out every item in a negative test), followed by retesting, but is there not an advantage to using information from the positive tests as well, to boost the performance of this design?

Overview

We thank the editor and reviewer for their thoughtful and constructive suggestions, which have helped us significantly strengthen the paper.

For your convenience, we have copied the review below (in *black italics*) with our responses provided point-by-point (in blue). References made (e.g., figure numbers) in the reviews were left as is; references in our responses correspond to the revised version.

We also indicate changes in the manuscript in blue.

To briefly highlight some key points of our revisions:

- Reviewer 4 Item 1 suggested restructuring the section “HYPER pooling method” to present the material in a more reader-friendly order.

We have restructured this section as suggested. We have also revised this section to condense the presentation and improve its clarity.

- Reviewer 4 Item 2 suggested removing a discussion of alternative strategies (that we don't employ) in order to explain the proposed method (HYPER) earlier in the paper.

The earlier version of our paper discussed these alternative strategies to illustrate the nontriviality of developing maximally balanced designs. We have moved the discussion to the **Methods**. As a result of this change (and other revisions made to reduce the overall length as requested by the editor), HYPER is now presented much earlier.

- Reviewer 4 Item 3 noted that **Fig 2** compares HYPER with alternative methods but does not include Dorfman testing and suggested including it.

We have added Dorfman testing to **Fig 2** and revised the corresponding discussion on **pgs 8-10**. Overall, Dorfman testing was more efficient than HYPER in the early days of epidemic growth but its efficiency substantially degraded as the prevalence rose.

- Reviewer 4 Item 4 asked about using more intricate decoding algorithms such as Definite Defective (DD) decoding.

We have added a new analysis (**Supp Fig 21**, discussed on **pg 16**) that considers using DD decoding. We consider two natural ways of incorporating it in HYPER, and find that they are either less effective or about as effective as the default HYPER (that uses conservative decoding).

We hope we have adequately addressed all the comments and look forward to hearing from you.

Response to Reviewer 4

This paper considers the problem of group testing (a.k.a. pooled testing). This is a method of making medical testing more efficient at a time of constrained test availability, by testing samples together in pools and (for example) ruling out all the samples in a pool which together tests negative. Due to the COVID pandemic, this is a particularly topical problem, and has been applied in a variety of circumstances. However, this has generally involved Dorfman-style pooling ('partition the population into distinct pools, if a pool is positive retest each member individually') and because of the parallel nature of PCR testing for example there is interest in designing test strategies which can proceed non-adaptively, or at least in a small number of stages, particularly in the presence of test error.

This paper achieves this by a test design based on so-called hypergraph factorization. To the best of my knowledge this is a novel strategy, and the simulations presented show that it performs well against a variety of standard test strategies. In that sense the work seems interesting and valuable and from looking at the past referees' reports I am inclined to say that the issues raised there have been properly dealt with. (For clarity, I should say that I have not reviewed previous rounds of submission of this paper. I was asked to take over from the previous Reviewer 2 and to provide a mathematical perspective on this work, but if my comments appear inconsistent with previous reviews or suddenly ask for things at a late stage then this is the reason).

Thank you very much for your positive and constructive comments and suggestions. They have helped us strengthen the paper and improve its presentation.

However, I have the following additional comments:

1. Although I am happy with the level of mathematical detail, I am not sure that it is explained in the right order. Specifically, on P6, the term 'hypergraph factorizations' is used, but this isn't defined until P9. I think it would make sense to restructure the material of P6-9, to first define what a hypergraph is (this may not be familiar to the general readership of this journal, particularly those from a medical background interested in group testing), then explain what is meant by a hypergraph factorization (ideally linked to an example), then explain how it is possible to use this to create a pooling strategy, and then to explain how the infection status of individuals can be deduced from this.

As suggested, we have restructured and revised this section ("HYPER pooling method").

The revised version presents the material in the following order:

1. **Setup:** Define what a hypergraph is and explain what is meant by hypergraph factorization (both linked to the example in **Fig 1**). Then describe how it produces the sequence of pool assignments (also linked to **Fig 1**).

2. **Stage 1:** Describe how the sequence of pool assignments is used to form pools and how the pooled testing results are decoded to identify putative positives (both linked to **Fig 1**).
3. **Stage 2:** Describe how putative positives are tested to declare positives (again linked to **Fig 1**).

Note that the term “hypergraph factorization” is now defined in the second paragraph.

Thank you for the suggestion!

2. Related, I'm not sure that it's worth saying (as on P5) that here's a strategy that we {em don't} employ -- i.e. cycling through pairs. Personally I'd prefer to explain the strategy (even in high level terms) that is used, as soon as possible in the paper.

In response to the editor's request to shorten the main text, we have moved this discussion (about the nontriviality of developing maximally balanced designs) to the **Methods** and replaced it with a high-level remark and pointer to the **Methods**.

3. While a comparison with alternative methods is made in Figure 2, and individual testing is a reasonable benchmark for this, I wonder whether it would be worth plotting the performance of standard Dorfman testing (with optimized pool size), at least in the noiseless case. That is, since the authors essentially consider a two-stage strategy (with conservative strategy of individual tests of those whose status is not clear), then it feels to me that Dorfman is a fair additional comparator. (Incidentally, it seems somewhat overcomplicated to plot the x axis of this as daily prevalence generated under an epidemic model -- it seems fine to me to plot simply prevalence directly, perhaps on a logarithmic scale).

We have added Dorfman testing (with matching pool size) to **Fig 2** and revised the discussion on **pgs 8-10**. Since Dorfman is a special case of HYPER (i.e., it is HYPER with $q=1$), we note the equivalence in the figure. Overall, the Dorfman designs were more efficient than the HYPER designs with $q=2$ in the early days of epidemic spread (i.e., prevalence below $\sim 1\%$ for the $n=96$ designs and below $\sim 0.2\%$ for the $n=384$ designs). However, the Dorfman designs lost significant efficiency as the prevalence grew; the $q=2$ HYPER designs were more efficient for higher prevalence.

Note that this behavior is consistent with our earlier comparisons under resource constraints, where design parameters were chosen to optimize the effective screening capacity. Early on, when the prevalence is low (e.g., as in **Supp Fig 11**), optimized Dorfman testing (i.e., HYPER with $q=1$) was more effective than HYPER with $q>1$ for a large set of constraints. When the prevalence is higher (e.g., as in **Supp Fig 16**), the optimized Dorfman testing is most effective over a much smaller set of constraints.

Regarding the x -axis, note that prevalence or log-prevalence would not be technically correct since the performance depends on the distribution of viral loads in the population, which is not

necessarily a function of prevalence alone. **Supp Fig 9** illustrates this point; e.g., days 82 and 104 have the roughly same prevalence but significantly different sensitivity.

4. In terms of decoding, even if it's not possible to prove anything formally, is there anything to be said by simulation about the performance of more intricate algorithms such as DD (in the language of [34]). It seems to me that this strategy is essentially COMP (rule out every item in a negative test), followed by retesting, but is there not an advantage to using information from the positive tests as well, to boost the performance of this design?

We have added a new experiment (**Supp Fig 21**, discussed on **pg 16**) that considers using DD decoding in HYPER.

In particular, we consider two ways of doing so:

1. **HYPER-DD-Decode:** Replace the conservative decoder in HYPER with the DD decoder.

In this scheme, the individuals passed on to stage 2 are the output of the DD decoder, i.e., only definite defective individuals are tested in stage 2. The rest are declared negative without further testing.

2. **HYPER-DD-Skip:** Use the DD decoder in conjunction with the conservative decoder to identify putative positives to skip in stage 2.

In this scheme, DD individuals are declared positive after stage 1 and do not undergo additional stage 2 testing. Non-DD putative positives still get tested in stage 2 as before.

Indeed, in both approaches, using DD has the potential advantage of using information from the positive tests to boost efficiency. Notably, the individuals that are tested in stage 2 are always a subset of the putative positives (who would all get tested in conservative decoding).

As expected, in our analysis, HYPER-DD-Decode retained efficiency as prevalence grew. However, it also lost significant sensitivity and was substantially less effective in testing-constrained settings.

HYPER-DD-Skip retains sensitivity (since non-DD putative positives are still tested) but it also did not significantly improve efficiency in our analysis. Consequently, its effective screening capacity was very similar to that of HYPER across the entire range of resource constraints.

Overall, neither approach yielded a significant improvement over HYPER here, and they both sacrifice some of the real-world benefits of conservative decoding (conceptual simplicity, only declaring positives from more well-understood individual testing, etc). As a result, we feel that conservative decoding continues to be a good choice in practice. Exploring further decoders may be an interesting direction for future research.